# Towards simplification of hydrologic modeling: identification of dominant processes

**S. L. Markstrom[1], L. E. Hay[1] and M. P. Clark[2]**

[1]{U.S. Geological Survey, PO Box 25046, MS 412, Denver Federal Center, Denver, Colorado, 80225, USA}

[2]{National Center for Atmospheric Research, P.O. Box 3000, Boulder, Colorado, 80307, USA}

Correspondence to: S. L. Markstrom (markstro@usgs.gov)

## Abstract

The Precipitation-Runoff Modeling System (PRMS), a distributed-parameter hydrologic model, has been applied to the conterminous United States (CONUS). Parameter sensitivity analysis was used to identify: (1) the sensitive input parameters and (2) particular model output variables that could be associated with dominant hydrologic process(es). Sensitivity values of 35 PRMS calibration parameters were computed using the Fourier Amplitude Sensitivity Test procedure on 110,000 independent hydrologically-based spatial modeling units covering the CONUS and then summarized to process (snowmelt, surface runoff, infiltration, soil moisture, evapotranspiration, interflow, baseflow, and runoff) and model performance statistic (mean, coefficient of variation, and autoregressive lag 1). Identified parameters and processes provide insight into model performance at the location of each unit and allow the modeler to identify the most dominant process on the basis of which processes are associated with the most sensitive parameters.

The results of this study indicate that: (1) the choice of performance statistic and output variables have a strong influence on parameter sensitivity, (2) the apparent model complexity to the modeler can be reduced by focusing on those processes that are associated with sensitive parameters and disregarding those that are not, (3) different processes require different numbers of parameters for simulation, and (4) some sensitive parameters influence only one hydrologic process, while others may influence many.

## 1 Introduction

It has long been recognized that distributed-parameter hydrology models (DPHMs) are complex because of the subtlety and diversity of the hydrologic cycle which they aim to simulate (Freeze and Harlan, 1969; Amorocho and Hart, 1964). In this study, two different aspects of this complexity are addressed:

(1) DPHMs have too many input parameters (Jakeman and Hornberger, 1993; Kirchner et al., 1996; Brun et al., 2001; Perrin et al., 2001; McDonnell et al., 2007). In this article, distributed parameters are defined as model inputs that remain constant through time, but can vary spatially across the landscape. Those who apply these models often have difficulty in the understanding of what these parameters are and how they are used in the model. Regularly, there are several parameters that may have similar effect on the computations or may constrain the model in unintended ways (Hrachowitz et al., 2014). Despite the

developer's claims that these DPHMs are more or less physically based, often there are not measurements or data sources available for reliable development of all of the input parameters. Duan et al. (2005) describes "a gap in our understanding of the links between model parameters and the land surface characteristics." These unmeasured parameters, ostensibly tangible, are really empirical coefficients when it comes to application and calibration (Samaniego et al., 2010).

(2) The output produced by DPHMs is difficult to interpret (Schaefli and Gupta et al., 2008; Gupta et al., 2009; Gupta et al., 2012; Mayer and Butler, 1993; Ewan, 2011). Often, the meaning of output variables is not always intuitive and results sometimes can seem contradictory (e.g. when streamflow does not seem to correlate with climate information). The result of these complex issues has led to the study of parameter interaction (Clark and Vrugt, 2006) and equifinality (Beven, 2006).

Developing effective DPHM applications require that the modeler address these two aspects of complexity at the same time (i.e. the uncertainty problem: "If I am uncertain when estimating input parameters, due to either incomplete or inaccurate information, what effect does it have on the output?", and the calibration problem: "I know the output I want, which parameters should I change and how much should I change them?") (Chaney et al., 2015; Reusser and Zehe, 2011). While, the user of a DPHM can do nothing about the complexity of the model's internal structure, the apparent complexity can be reduced by limiting the parameters and the affected output under consideration (as described by Jakeman and Hornberger, 1993; Hay et al., 2006).

Global parameter sensitivity analysis can determine the degree to which different values of parameters can affect the simulation of certain model outputs (Sanadhya et al., 2013). Furthermore, parameter sensitivity can be evaluated with respect to selected output variables, each representing a different aspect of the hydrologic cycle (hereafter referred to as *processes*). Sensitivity analysis of this form can be used to both identify the input parameters that are the most sensitive (i.e. the parameters that affect the simulation the most) and the dominant process(es) (i.e. those processes which are affected most, by the most sensitive parameters) according to the DPHM.

Results of parameter sensitivity analysis can vary spatially (van Werkhoven et al., 2008). Certain parameters can be more or less sensitive at different locations on the landscape. For example, parameters related to simulation of snow can become more sensitive at higher

elevations, while parameters related to evaporation can become less sensitive at locations
where soil depth and the overall capacity for soil water storage decreases. Consequently, the
dominant process(es), as identified by parameter sensitivity analysis of the DPHM, will vary
across the landscape as well.
Any particular DPHM must necessarily be able to simulate any and all hydrological processes
that may occur anywhere on the landscape. However, with the application of a DPHM to a
specific site, it can become much less complex when the dominant hydrological process(es)
are identified, as not all processes are active to the same degree. The modeling problem
becomes less complex to the modeler when hydrological processes not relevant to the
modeled domain or watershed are removed from consideration (Wagener et al., 2003; Reusser
et al., 2011; Guse et al., 2014; Bock et al., 2015). Related to this, various methods have been
developed that will group similar watersheds together for purposes of study (Wolock et al.,
2004; Winter, 2001; Ali et al., 2012) or for parameter regionalization (He et al., 2011; Merz
and Blöschl, 2004, Seibert, 1999; Vogel 2005). In addition, dominant process concepts have
been explored as a way to classify watersheds and natural hydrologic systems for the purpose
of simplifying DPHMs by several researchers (Sivakumar and Singh, 2012; Sivakumar et al.,
2007). Some have suggested this approach for use as a possible classification framework
(e.g. Woods, 2002; Sivakumar, 2004). Pfannerstill et al. (2015) developed a framework for
identification and verification of hydrologic process in simulation models on the basis of
temporal sensitivity analysis. Cuntz et al. (2015) describe a method of identifying only
informative parameters as a screening step in order to reduce the effort required to perform
global sensitivity analysis on the full parameter space. McDonnell et al. (2007) discuss the
possibility of simplifying hydrologic modeling by identifying "fundamental laws" so that
overparameterized models are not needed. However, in our opinion we have not made much
progress on that front and DPHMs are, in many ways and for many reasons, more complex
than ever.
This article describes an approach for identification of sensitive parameters and processes for
a modeling application of the conterminous United States (CONUS, Fig. 1). Identification
and simulation of regional CONUS sub-watersheds are determined by the resolution of the
available information and how the DPHM responds to geophysical (e.g., topography,
vegetation and soils) and climatological variation. Specifically, we propose to identify the
sensitive parameters and dominant hydrologic process(es), thereby reducing the amount of
parameter input and number of output variables to consider (Chaney et al., 2015) and address
the two aspects of complexity as described above.

## 2    Methods

### 2.1    Distributed-parameter hydrology model

The U.S. Geological Survey's Precipitation-Runoff Modeling System (PRMS) is the DPHM
used in this study.   PRMS is a modular, deterministic, distributed-parameter, physical-process
watershed model used to simulate and evaluate the effects of various combinations of
precipitation, climate, and land use on watershed response.    Each hydrologic process
simulated by PRMS is encoded in a modular piece of source code (i.e. a "module") and is
represented by an algorithm that is based on a physical law (i.e. balance of energy required to
melt the ice in a snowpack) or empirical relation with measured or estimated characteristics
(i.e. a tank model used to simulate interflow).    The reader is referred to Markstrom et al.
(2015) for a complete description of PRMS.
A fundamental assumption of this study is that PRMS is able to simulate and differentiate
hydrologic signals from all the different processes at the scale of the CONUS.   Two possible
ways to evaluate this are: (1) an analysis of PRMS's internal structure, and (2) the history of
PRMS applications.    A detailed analysis of PRMS's structure is beyond the scope of this
article (see Markstrom et al., 2015); however, PRMS is implemented in a very linear fashion.
Each parameter is clearly identified with an equation that is related to simulation of a specific
process.    Equations are solved sequentially, generally in the order that is defined by water
moving through the hydrologic cycle, starting from the atmosphere as precipitation and
moving through the rivers as streamflow.    The outputs of one equation may be used as inputs
to subsequent equations.    All of the inputs for a particular equation are required before that
equation can be solved.    This interdependency in equations can lead to parameter interaction
in the simulation of subsequent processes (as described by Beven, 1989; Grayson et al., 1992;
Yilmaz et al., 2008; Pfannerstill et al., 2015).    For example, parameters related to distribution
of temperature and solar radiation may show correlation with each other when evaluated with
respect to simulation of evapotranspiration despite these parameters not being explicit terms
in the evapotranspiration equations.    Past studies indicate that PRMS has been very useful in
water-resource and research studies across the CONUS (Battaglin et al., 2011; Boyle et al.,
2006; Hay et al., 2011; Markstrom et al., 2012) and is capable of matching measured data
(Bower, 1985; Cary, 1991; Dudley, 2008; Koczot et al., 2011) in a variety of geophysical and
climatological settings.
To define the spatial domain for the CONUS application of PRMS, the locations of major
river confluences, water bodies, and stream gages have been georeferenced. Approximately
56,000 stream segments are used to connect these locations. Using these stream segments,
the left and right bank areas that contribute runoff directly to each segment have been
identified, resulting in approximately 110,000 irregularly shaped hydrologic response units
(HRUs) of various sizes (500 $m^2$ to 14,000 $km^2$) (Viger and Bock, 2014). These HRUs are
derived by their geographic and topographic location, affecting their extent and resolution.
The CONUS application is forced with values of daily precipitation and daily maximum and
minimum air temperature from the DAYMET data set (Thornton et al., 2014). The climate
information covers a time period from 1980-2013 on a daily time step, but a shorter period
(1987 – 1989 used for warmup, and 1990 – 2000 used for evaluation) was used in this study.

## 2.2   Calibration parameters

The version of PRMS used in this study has 108 input parameters. A parameter is defined as
an input value that does not change over the course of a simulation run. Of these parameters,
most would never be modified from their initial values (hereafter referred to as *non-*
*calibration parameters*, see Viger, 2014) because they are (1) computed directly from digital
data sets through the use of a geographic information system (e.g. land-surface
characterization parameters), (2) boundary conditions (e.g. parameters to adjust daily
precipitation and daily air temperature forcings), or (3) model configuration options (e.g. unit
conversions and model output options). This leaves 35 parameters under consideration for
improved model performance, hereafter referred to as *calibration parameters* (Table 1). Each
parameter is used within a PRMS code module that simulates a single hydrologic process in
PRMS. The output variables of one module may be used as input variables to other modules.
It is through these connections that calibration parameters associated with a PRMS module
may affect the results of other modules.

## 2.3   Hydrologic processes

PRMS produces more than 200 output variables that indicate the simulated hydrologic
response of a watershed through time (Markstrom et al., 2015, see Table 1-5). In this study,

eight of these output variables have been selected to represent the response of major hydrologic processes at the HRU resolution. These processes are: (1) snowmelt (PRMS output variable *snowmelt*) – the amount of water that has changed from ice to liquid and becomes either surface runoff or infiltrates into the soil zone of the HRU; (2) surface runoff (*sroff*) – water from a rainfall or snowmelt event that travels quickly over the land surface from the HRU to the connected stream segment; (3) infiltration (*infil*) – the sum of rain and snowmelt that passes into the soil zone of the HRU; (4) soil moisture (*soil_moist*) – the storage state that represents the amount of soil water in the soil zone above wilting point and below total saturation in the HRU; (5) evapotranspiration (*hru_actet*) – the total actual evapotranspiration lost from canopy interception, snow sublimation, and soil and plant losses from the root zone; (6) interflow (*ssres_flow*) – shallow lateral flow in the unsaturated zone to the connected stream segment; (7) baseflow (*gwres_flow*) – the component of flow from the saturated zone to the connected stream segment; and (8) runoff (*hru_outflow*) – the total flow from the HRU contributing to streamflow in the connected stream segment. It is assumed that these eight output variables are representative of the processes typically considered in hydrological studies with DPHMs. Details of how these processes are simulated by PRMS are described by Markstrom et al. (2015).

## 2.4   Performance statistics

For DPHMs, there are many different performance measures that have been developed for different purposes (Krause et al., 2005; Gupta et al., 2008; Gupta et al., 2009; Mendoza et al., 2015a; Mendoza et al., 2015b). Because this study is an analysis of model sensitivity, the performance measures need only track changes in model output and do not necessarily need to include observed measurements. Consequently, performance statistics can be developed for processes that are not normally evaluated by performance measures. Archfield et al. (2014) demonstrated that seven fundamental daily streamflow statistics (FDSS) can be used to group streams by similar hydrologic response and tend to provide non-redundant information. In this study, all seven FDSS were computed for each of the eight PRMS time series output variables corresponding to the processes. For the purpose of illustration, this article focuses on three of the FDSS: (1) mean; (2) coefficient of variation (CV); and (3) the autoregressive lag-one correlation coefficient (AR-1). In an intuitive sense, these three statistics can be thought to represent changes in total volume, "spikiness" or "flashiness", and day-to-day

timing, respectively. These performance statistics are computed on the daily time series of the process variables for the 10-year evaluation period.

## 2.5 FAST analysis

Parameter sensitivity analysis measures the variability of model output given variability of calibration parameter values. This is determined by partitioning the total variability in the model output or change in performance statistics to individual calibration parameters (Reusser et al., 2011). The Fourier Amplitude Sensitivity Test (FAST) (Schaibly and Shuler, 1973; Cukier et al., 1973; Cukier et al., 1975; Saltelli et al., 2006) was selected for this study because it has been demonstrated that it can efficiently estimate non-linear hydrologic model parameter sensitivity (Guse et al., 2014; Pfannerstill et al., 2015; Reusser et al., 2011). FAST is a variance-based global sensitivity algorithm that estimates the first-order partial variance of model output explained by each calibration parameter (hereafter referred to as *parameter sensitivity*). Specifically, this first-order variance is the variability in the output that is directly attributable to variations in any one parameter and is distinguishable from higher order variances associated with parameter interactions. An important caveat is that these higher order variances are not accounted for in the analysis. It is assumed that first-order partial variance is sufficient to identify sensitive parameters. This same assumption, as applied to process identification, may be more problematic. If there are sets of interactive sensitive parameters that have not been identified, then the associated process(es) will not be identified as such.

Selected parameters are varied within defined ranges at independent frequencies among different model runs. FAST identifies the variability of parameter sensitivities and their ranks, by means of their contribution to total power in the power spectrum. FAST has been implemented as the 'fast' library in the statistical software R (Reusser et al., 2011; Reusser, 2013; R Core Team, 2015) in two parts. In the first part, the user identifies the calibration parameters and respective value ranges for the test, then FAST generates sets of test calibration parameter values (hereafter referred to as *trials*). Calibration parameter values are varied across the trials according to non-harmonic fundamental frequencies. The user then runs the DPHM for each trial and computes corresponding performance statistics. Then the user runs the second part of the FAST package that performs a Fourier analysis of the performance statistics over the trial space looking for the frequency signatures associated with each calibration parameter.

The FAST methodology results in a simple procedure for computing parameter sensitivities on an HRU basis for all the CONUS. The steps in this process are as follows:

1. Assign appropriate ranges for the 35 calibration parameters (Markstrom et al., 2015; as in LaFontaine et al., 2013).

2. Run the first part of the FAST procedure (as described above) to develop over 9000 unique parameter sets, comprised of value combinations for the calibration parameters. The total number and content of these parameter sets, and the results from their simulation by PRMS are completely determined by the first part of the FAST procedure in order to investigate the trial space. Each of the prescribed simulations are independent of each other so they can run in parallel on a computer cluster.

3. Compute the FDSS based performance statistics (mean, CV, and AR-1) for each process.

4. Run the second part of the FAST procedure (as described above) using output from step 3, resulting in PRMS parameter sensitivities, at each HRU, for the 56 combinations of seven performance statistics and eight processes (plus totals).

## 3  Results

### 3.1  Parameter sensitivity by process and performance statistic

Figure 2 shows parameter sensitivity as a set of maps ordered by process and performance statistic. This illustrates the spatial variability in parameter sensitivity and the importance that choice of performance statistic can make in terms of evaluation of hydrologic response. In these maps, the HRUs are colored according to the parameter sensitivity, which is computed by summing the first-order sensitivity for all 35 parameters separately for each of the 8 output variables, each corresponding to their respective process. These sums do not necessarily sum to one, and then scaling each individual category of modeled process and performance statistic to total sensitivity. This summed sensitivity across the parameters, by each category is hereafter referred to as *cumulative parameter sensitivity*. Parameter sensitivity associated with process (column labeled "Process average" in Figure 2) are averaged across all of the parameter sensitivity values computed for the different performance statistics, while parameter sensitivity associated with the performance statistics (last row labeled "Performance statistic average" in Figure 2) are averaged across all of the parameter

sensitivity values computed for the different processes. These categories are indicated by
their position in the rows and columns in Figure 2. When looking at a single performance
statistic for a single process, the cumulative parameter sensitivity can vary from near 0.0
(white colored HRUs) to near 1.0 (black colored HRUs). Low values in these maps indicate
that there are no parameters that can be changed in any way to affect the performance statistic
(this situation is hereafter referred to as an *inferior process*). Likewise, each HRU has a
cumulative sensitivity value (i.e. the sum of all of the partial sensitivities for each process).
The process with the largest sum on an HRU is referred to as the *dominant process* for that
HRU.
An example of an inferior process is clearly seen in the case of the mean of the snowmelt
process in the southern CONUS HRUs. This is because the occurrence of snow in these areas
is very infrequent. Also, there were HRUs for which the value of some performance statistics
were mathematically undefined for certain processes (e.g. AR-1 and CV for the baseflow and
snowmelt processes). These cases occur when the output variable representing the process
does not change at all through time, regardless of the parameter values, and are extreme
examples of inferior processes. Likewise, a clear example of a dominant hydrologic process
is the CV of interflow in the Intermountain West region of the CONUS (Figs. 1 and 2). This
means that for these HRUs, there exist some calibration parameters that can be varied that
affect this process to a very high degree.
Also apparent from Figure 2 is that there are clear spatial patterns in the parameter sensitivity
on the basis of the geographical features of the CONUS. Generally, many of the maps show
a sharp break in parameter sensitivity between mountain ranges and comparatively lower
elevations, northern contrasted with southern latitudes, and humid versus arid climates.
Specific contrasts can be seen in several maps such as when examining the Humid Midwest
as opposed to the Great Plains regions and the Pacific Coastal areas and the Desert Southwest
region of the CONUS (Fig. 1). Additionally, topographic features of the landscape are
prominent (e.g. elevation for interflow), while in other maps, climate considerations seem to
dominate (e.g. snowmelt). Another specific example is that the mean of each process, which
indicates the ability of any parameter(s) to change the total volume of water during a
simulation, seems to have a low sensitivity band in the Great Plains region for all processes
except for snowmelt (Fig. 1). This band of low sensitivity has been noted in other modeling
studies (Newman et al., 2015; Bock et al., 2015).

## 3.2 Parameter count required to parameterize each process

To identify the expected count of parameters required to parameterize a particular process, cumulative parameter sensitivity across all HRUs of the CONUS has been computed and plotted (Fig. 3(a)—(h)). The sensitivity level accounted for by the most sensitive parameter, regardless of which parameter it is, for all HRUs across the CONUS is plotted in position 1 on the X axis of each of these plots (Fig. 3(a)—(h)). Then, cumulative sensitivity is plotted for the parameter in rank 2, and so on, until the cumulative sensitivity of all 35 calibration parameters is accounted for. The plots in Figure 3(a)—(h) show that far fewer than the full 35 parameters are needed to account for most of the parameter sensitivity. In fact, to account for 90% of the parameter sensitivity, this count varies from a low value of just over two for snowmelt to an average high value of over 9 for runoff in selected HRUs.

The actual count of calibration parameters required to account for 90% of the parameter sensitivity varies by process and region, as shown by the maps in Figure 3(i)—(p). These maps were generated by counting the number of parameters required to obtain the 90% cumulative sensitivity level for each HRU. For example, Figure 3(o) indicates that for the baseflow process between three and nine parameters are needed to account for 90% of the parameter sensitivity in the various HRUs across the CONUS, with the higher count needed in mountainous, Great Lakes, and New England regions. The maps also indicate that between two (Fig. 3(i)) to 13 parameters (Fig. 3(k, n, and p)) are required for parameterization of these processes. This analysis indicates that more parameters are needed to simulate the components of streamflow (e.g. baseflow, interflow, and surface runoff) than processes that do not result directly in flow (e.g. snowmelt, evapotranspiration, and soil moisture). In addition, simulated processes that are identified as being sensitive to parameters with which they are not normally associated with, may indicate that these processes are a convolution of other processes, consequently making parameters sensitive that are not normally sensitive.

Visually, these maps (Fig. 3(i)—(p)) indicate that HRU calibration parameter counts vary regionally. For most processes, higher parameter counts are seen in the more mountainous regions of the Cascade, Sierra Nevada, Rocky, Ozark, and Appalachian mountains, although this is true to a much lesser extent for the evapotranspiration and soil moisture processes (Figs. 3(m) and 3(l)). Higher values also seem prevalent in the New England and Great Lake regions (Fig. 1). This result seems to indicate that, no matter which part of the hydrologic

cycle is simulated, more parameters are required in these regions. In contrast, low parameters
counts seem prevalent in the Great Plains and Desert Southwest regions.
Finally, Figure 3 illustrates the extent to which it is possible to decompose the parameter
estimation problem into a sub-set of independent problems, and hence reduce the
dimensionality of the inference problem and avoid the troublesome nature of parameter
interactions. By considering a single (or reduced set of) processes and performance statistic
categories at a time, the sensitive parameter space can be substantially reduced. It also
illustrates that there is a strong spatial component to this decomposition. In order to make the
information presented in Figure 3 more useful for DPHM application, the particular sensitive
parameters have been determined for each HRU by ranking the calibration parameters by
sensitivity for each category of process and performance statistic for each individual HRU
and is summarized by counting the occurrence of each parameter across the HRUs and
ranking them within their respective category of process and performance statistic (Table 2).
To address the issue of the spatial variability of these parameters, the percentage of the total
number of HRUs for which that parameter is sensitive is shown as the number in parentheses
after the parameter name in Table 2. Higher percentage values would indicate that the
corresponding parameter is sensitive across more of the CONUS. Refer to Table 1 for a
complete description of these parameters.
When looking at the categorical parameter lists of Table 2, it is expected that different
parameters would associate with different processes (i.e. along a column), but it is surprising
to see how different the parameter lists are for different performance statistics (moving across
a row) for the same process. An example of this is the baseflow process: the baseflow
coefficient (PRMS parameter *gwflow_coef*) is the most sensitive parameter for performance
statistics s CV and AR1, but is not even in the list of sensitive parameters for the performance
statistic related to the mean of the process. This implies that this parameter is influential for
affecting the timing of baseflow, while it does not have any effect on the total volume of
baseflow.
Further inspection of Table 2 indicates that some calibration parameters occur in many of the
24 categories (8 processes times 3 performance statistics), while some parameters do not
occur at all. A count of how many times each parameter occurs provides insight into how
many process/performance statistic combinations that particular parameter influences. To
investigate this for the CONUS application, another view of the information in Table 2 is

shown in Figure 4. The 25 sensitive calibration parameters from Table 2 are listed on the y-axis of Figure 4, ranked by order of the number of times that they appear in the process/performance statistic categories. Furthermore, each appearance is indicated by an adjacent circle. Independent of the number of times a parameter occurs within a category (number of circles), the color of the circle visually indicates the proportion of the CONUS HRUs that are affected by that parameter. Specifically, a red circle indicates that more HRUs are affected, while blue indicates that fewer HRUs are affected.

Figure 4 shows that three specific parameters affect 18 or more process/performance statistic categories; seven parameters affect seven to 14 categories, and 15 specific parameters affect one to five categories. Finally, of the 35 parameters studied, 10 are never used for any combination of process and performance statistic (Table 2 and Fig. 4). It is apparent from Figure 4, that for the CONUS application of PRMS, the parameters affecting the most process categories are *soil_moist_max* (maximum available water holding capacity), *jh_coef* (Jensen-Haise air temperature coefficient), and *dday_intcp* (intercept in degree-day solar radiation equation). Because these parameters affect so many categories, modelers would be wise to invest their resources in developing the best values possible for these parameters to avoid unintended parameter interaction during calibration. Ideally, these parameters could be estimated from reliable external data and set for the model and not calibrated. The parameters that affect the least number of process categories (aside from the parameters that are never sensitive) are *cecn_coef* (convection condensation energy coefficient), *ssr2gw_exp* (coefficient in equation used to route water from the soil to the groundwater reservoir), *emis_noppt* (emissivity of air on days without precipitation), *potet_sublim* (fraction of potential evapotranspiration that is sublimated), and *slowcoef_lin* (slow interflow routing coefficient). Ideally, these parameters could be set to default values since there is limited value in calibrating them.

Also apparent from Figure 4 is that there are many parameters between these two extreme groups. Parameters like *smidx_coef* (soil moisture index for contributing area calculation) can appear in several process categories, without any high rankings, while there are other parameters like *slowcoef_sq* (slow interflow routing coefficient) that appear in relatively few process categories, but have high rankings. This behavior may be due to the vertical routing order (i.e. processes that occur nearer to the surface happen before the deeper ones) of the associated processes (Yilmaz et al., 2008; Pfannerstill et al., 2015). In PRMS, the process of

partitioning of precipitation into either direct surface runoff or infiltration (controlled directly by parameter *smidx_coef*) is "faster" and occurs in the vertical routing order before the process of interflow generation (controlled directly by parameter *slowcoef_sq*). These parameters may be the best candidates for calibration because they are sensitive, while at the same time interaction across processes is perhaps limited.

### 3.3  Identification of dominant and inferior processes by HRU

To identify the dominant and inferior process(es) by geographic area, the following procedure is done for each HRU:

1. The parameter sensitivity scores are summed for each parameter, resulting in a score for each parameter for each time series output variable and performance statistic.

2. The parameter scores are averaged by performance statistics, resulting in a score for each process.

3. The process scores are ranked for each HRU.

4. The top (and bottom) ranked process determines the most dominant (and most inferior) single process for each HRU as shown in Figure 5.

Generally, Figure 5(a) shows that evapotranspiration is the most prevalent dominant process for the CONUS. This is probably because it is a major component of the hydrologic cycle and sensitive parameters are available to affect it in every HRU. However, this is not universal, and the dominant process varies by geographic region, with snowmelt being the dominant process in the northern Great Planes and northern Rocky Mountains, total runoff being the most important in the Pacific Northwest, and with interflow important in bands across the Intermountain West (Fig. 1). Each process is dominant somewhere depending on local conditions. Equally informative are the locations of the most inferior processes (Fig. 5(b)). This clearly shows that PRMS snowmelt parameters are not sensitive across the Central Valley of California, and in the Deep South and the Southwestern United States (Fig. 1). Areas where runoff is more dominant than evapotranspiration, as in the Cascade Mountains and coastal areas of the Pacific Northwest, are locations where the runoff is a substantially greater part of the water budget. Interestingly, infiltration and baseflow appear to be equally inferior across most of CONUS, with pockets of HRUs that are insensitive to

soil moisture, surface runoff, and interflow, depending on local conditions. There are no HRUs that rank evapotranspiration as the most inferior process.

Dominant and inferior processes can be identified for HRUs at the watershed scale as well. Figure 5(c) shows the most dominant process by HRU for the Apalachicola – Chattahoochee– Flint River watershed in the Southeastern United States. This watershed has been the subject of previous PRMS modeling studies (LaFontaine et al. 2013). When using this information at a finer resolution, it shows that evapotranspiration is the most dominant process watershed wide, but with pockets of HRUs in the northern part of the watershed where runoff is the most dominant and a pocket in the southern part of the watershed where infiltration is most dominant. Likewise, the most inferior process for each HRU is identified in Figure 5(d). This clearly indicates that parameters and performance statistics related to snowmelt, and to a lesser degree baseflow do not need to be considered when modeling this watershed. Figure 5(d) also indicates, that in the northern part of the watershed, infiltration and runoff are inferior processes as well, which could in part be due to impervious conditions around the Atlanta metropolitan area.

## 4   Discussion

### 4.1   Causes of parameter sensitivity

There are regions where parameter sensitivity is typically high for a particular performance statistic (e.g. New England region [Fig. 1] for performance statistic based on mean of processes) or typically low (e.g. Great Plains region [Fig. 1] for mean of processes) regardless of the process (Fig. 2). Why do the HRUs of some regions exhibit parameter sensitivity to almost all processes, while others exhibit parameter sensitivity to almost none? All other things being equal, there can only be two sources of these spatial patterns:

1. The physiography that is used to define the non-calibration parameters (e.g. elevation, vegetation type, soil type) renders all calibration parameters insensitive. A theoretical example of this could be if an HRU is characterized as entirely impervious, resulting in the non-existence of any simulated soil water.

2. Patterns in the climate data used to drive the model (e.g. daily temperature and precipitation) could control model response. A theoretical example of this could be an HRU that receives no precipitation. The hydrologic response of the HRUs in either case would always remain unchanged, regardless of changes in any parameter value.

In either case, these sources of information are independent of the DPHM and could lead to
the conclusion that the dominant processes identified by the methods outlined in this article
could correspond to perceptible dominant processes in the physical world (i.e. how the "real
world" works).
The number of unique calibration parameters for each process in Table 2 (i.e. counting the
parameters across each row) may provide some insight into the complexity of each process as
represented in the model structure of PRMS. In theory, more "complicated" hydrologic
processes would require more parameters for parameterization than the "simpler" ones.
According to this view, runoff (16 calibration parameters), infiltration (12 calibration
parameters), and interflow (12 calibration parameters) are the most complex processes to
simulate, with soil moisture (4) being the simplest. Baseflow (11 calibration parameters),
snowmelt (11 calibration parameters), surface runoff (10 calibration parameters), and
evapotranspiration (8 calibration parameters) are in between. This reflects the fact that in
PRMS, runoff is a much more complicated calculation with many of the other processes
directly contributing information. Also apparent is that more parameters are needed to
simulate the components of streamflow (e.g. baseflow, interflow, and surface runoff) than
processes that do not result directly in flow (e.g. snowmelt, evapotranspiration, and soil
moisture). The only process that does not follow this pattern is infiltration. Storm-event
based infiltration is typically simulated with sub-daily time steps to account for the
time/intensity variability of this process. It is possible that PRMS must compensate for this
shortcoming in structure with a more complex parameterization of the process.
Table 2 indicates that there are 10 calibration parameters that are never sensitive regardless of
the process or performance statistic. This indicates that these parameters should always be set
to the default value, with minimal resources used to estimate them, and never be calibrated.
Additional modeling studies could reveal situations where these parameters actually do
exhibit some sensitivity, perhaps in situations with smaller geographical domains or over
different time periods. It is also possible that these parameters are never sensitive, indicating
some structural problem or unwarranted complexity in the DPHM and the removal of some
algorithms from the source code of the DPHM is advised. Additional study is required of
these 10 non-sensitive calibration parameters and upon further review of the PRMS source
code, a structural problem (e.g. unintended constraint, non-differentiable behavior, or
software bug) might be revealed. Alternatively, the problem could be related to invalid
parameter ranges in the FAST analysis or problems with the climate data used to drive the
model. Finally, it could be that alternative or improved performance statistics could resolve
this issue.

## 4.2 Choice of performance statistic

The maps of Figure 2 clearly illustrate the importance that choice of performance statistic can
make in terms of evaluation of hydrologic response. When the maps of performance statistics
within a single hydrologic process are compared (i.e. the maps across a single row), the
spatial patterns and magnitude of the parameter sensitivity can be very different. This could
indicate that the performance statistics based on the FDSS truly are non-redundant and are
accounting for different aspects of the processes.
Table 2 indicates that the baseflow coefficient (PRMS parameter *gwflow_coef*, Markstrom et
al., 2015) is the most sensitive parameter for performance statistics CV and AR1, but not
sensitive to the mean of the baseflow process performance statistics. This points to the fact
that despite having knowledge of a parameter being associated with the computation of a
certain process, sensitivity analysis can reveal that the response of the simulation is
completely different when the performance statistic changes. It also indicates that sensitivity
analysis might be an important step in selection of an appropriate performance statistic and
that uncritical application of performance statistics may be misleading.

## 4.3 Spatial aspects of dominant and inferior processes

When the dominant and inferior processes are determined for an HRU (Fig. 5), it is possible
that certain parameters are included in both the most dominant dominate and most inferior
processes at the same time. This apparent contradiction is not necessarily a conflict but
indicates that the calibration parameters must work in concert with the evaluation method.
For example, there exist HRUs where the evapotranspiration process is dominant and at the
same time the runoff or infiltration processes are inferior (Fig. 5(a) and 5(b)). The parameter
*soil_moist_max* is indicated as being sensitive for all three of these processes (Table 2). This
parameter would demonstrate equifinality if evaluated within the context of the inferior
processes (i.e. those output variables and performance statistics associated with the inferior
process) but would be a very effective calibration parameter resulting in optimal values when
viewed within the context of parameters and variables of the dominant process.
This method of identification of inferior and dominant processes for a specific geographical
location (i.e. HRU, watershed, or region), determined by sensitivity analysis, is defined within
the context of the application of the DPHM and may not necessarily have the same meaning
within a different context. However, this methodology does have the ability to spatially
classify watersheds and identify dominant processes. This classification scheme depends not
only on the physiographic nature of the watershed, but also on the scale, resolution, and
purpose that were considered by the modeler when the application was developed.
## 4.4  Further study
Providing modelers with reduced lists of calibration parameters on an HRU-by-HRU,
watershed-by-watershed, or region-by-region basis is the first step in the path of this research.
Subsequent steps to this approach could be developed into more sophisticated methods where
orthogonal output variables and performance statistics could provide much more insight into
methods of effective model calibration. Other advancements in this approach may identify
groups of parameters that effectively behave together, thus reducing the number of parameters
and making specific model output respond more directly to a single or a few parameters,
reducing parameter interaction. This suggests that model parameterization and calibration
might benefit from a step-by-step strategy, using as much information as possible to set non-
interactive parameters and remove them from consideration before the more interactive
parameters are calibrated, reducing the dimensionality of the problem (Hay et al., 2006; Hay
and Umemoto, 2006).
Another question for future research is: Does the classification of dominant hydrologic
processes, both geographical and categorical, as described in this study, apply in other
contexts? Comparable findings from other modeling studies, such as those by Newman et al.
(2015) and Bock et al. (2015), might indicate that there could be a connection. These other
studies use the same input information (i.e. being driven with the same climate data and using
the same sources of information for parameter estimation), and thus simulation results and
model sensitivity to this information might be similar. Also, can real world watersheds be
classified by sensitivity analysis using DPHMs? Based on the findings of the work presented
so far, the answer is inconclusive. Clearly there are some results that indicate that it might be
possible. For example, the methods described here effectively identify "snowmelt
watersheds" in the mountainous and northern latitudes, but, is all of this necessary to
accomplish this? Might simpler methods (e.g. an isohyetal snowfall map) identify snowmelt
watersheds just as effectively?
Questions remain about using parameter sensitivity for identification of structural
inadequacies within the CONUS application and specifically the PRMS model itself. A full
analysis of these parameters and how they relate to their respective process(es) is beyond the
scope of this article, but it could relate information about the structure of PRMS. In this
study, certain hydrologic processes (e.g. depression storage, streamflow routing, flow through
lakes, and strong groundwater/surface-water interaction) were not considered because of
additional data requirements and parameterization complexity. The PRMS model also allows
for selection of alternative methods for many of the module types. Each of these modules
uses different equations and calibration parameters. Future work might be to determine the
effect of using different modules or maybe even to determine the selection of the PRMS
modules through sensitivity analysis. Just as the spatial and temporal scope of any modeling
project must be defined, the scope of the hydrologic processes, and the detail to which these
processes are simulated, must be likewise defined. Also, alternative ways of defining HRUs
(e.g. larger or smaller, or even based on dominant process instead of geographic location)
could affect the analysis. Model development and application could perhaps proceed by first
accounting for those factors that have the most effect.

## 5 Conclusion

Watersheds in the real world clearly exhibit hydrologic behavior determined by dominant
processes based on geographic location (i.e. land surface conditions and climate forcings). A
methodology has been developed to identify regions, watersheds, and HRUs according to
dominant process(es) on the basis of parameter sensitivity response with respect to a
distributed-parameter hydrology model. The parameters in this model were divided into two
groups – those that are used for model calibration and those that were not. A global
parameter sensitivity analysis was performed on the calibration parameters for all HRUs
derived for the conterminous United States. Categories of parameter sensitivity were
developed in various ways, on the basis of geographic location, hydrologic process, and
model response. Visualization of these categories provides insight into model performance,
and useful information about how to structure the modeling application should take advantage
of as much local information as possible.

By definition, an insensitive parameter is one that does not affect the output. Ideally, a distributed-parameter hydrology model would have just a few calibration parameters, all of them meaningful, each controlling the algorithms related to the corresponding process. This would result in low parameter interaction and a clear correspondence between input and output. However, this is not always the case, and despite the fact that parameter interaction is unavoidable in these types of models, this behavior is also seen in the real world. For instance, in watersheds where evaporation is very high, antecedent soil moisture is affected, which has a direct influence on infiltration. The real world process of evaporation has an effect on infiltration, just as evaporation parameters have an effect on simulation of infiltration in watershed hydrology models. Application of distributed-parameter hydrologic modeling application require that the uncertainty problem and the calibration problem be addressed at the same time. While, the user of a DPHM can do nothing about the complexity of the model's internal structure, the apparent complexity can be reduced by limiting the parameters and the affected output under consideration.

Results of this study indicate that it is possible to identify the influence of different hydrologic processes when simulating with a distributed-parameter hydrology model on the basis of parameter sensitivity analysis. Factors influencing this analysis include geographic area, topography, land cover, soil, geology, climate, and other unidentified physical effects. Identification of these processes allows the modeler to focus on the more important aspects of the model input and output, which can simplify all facets of the hydrologic modeling application.

## Data availability

The Precipitation-Runoff Modeling System software used in this study is developed, documented, and distributed by the U.S. Geological Survey. It is in the public domain and freely available from their web site (http://wwwbrr.cr.usgs.gov/prms). Data analysis and plotting is done with the R software package (http://www.r-project.org), which is freely available, subject to the GNU General Public License.

The climate forcing data set used in this study came from the U.S. Geological Survey Geo Data Portal (http://cida.usgs.gov/climate/gdp). The HRU delineation and default parameterization came from the U.S. Geological Survey GeoSpatial Fabric (http://wwwbrr.cr.usgs.gov/projects/SW_MoWS/GeospatialFabric.html). Finally, the parameter sensitivity output values that were used to make the maps and tables in this article are available at ftp://brrftp.cr.usgs.gov/pub/markstro/hess.

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

**Tables**
Table 1. Precipitation Runoff Modeling System (PRMS) calibration parameters used in this
study. The values in the column labeled "PRMS module" identify the module type equation(s)
from the PRMS source code (see Markstrom et al., 2015).

| Parameter name | Description | PRMS module | Range |
|---|---|---|---|
| adjmix_rain | Factor to adjust rain proportion in a mixed rain/snow event | climate | 0.6—1.4 |
| tmax_allrain | Maximum air temperature above which precipitation is rain | climate | -8.0—60.0 |
| tmax_allsnow | Maximum air temperature below which precipitation is snow | climate | -10.0—40.0 |
| dday_intcp | Intercept in degree-day equation | solar radiation | -60.0—10.0 |
| dday_slope | Slope in degree-day equation | solar radiation | 0.2—0.9 |
| ppt_rad_adj | Solar radiation adjustment threshold for precipitation days | solar radiation | 0.0—0.5 |
| radj_sppt | Solar radiation adjustment on summer precipitation days | solar radiation | 0.0—1.0 |
| radj_wppt | Solar radiation adjustment on winter precipitation days | solar radiation | 0.0—1.0 |
| radmax | Maximum solar radiation due to atmospheric effects | solar radiation | 0.1—1.0 |
| tmax_index | Temperature to determine precipitation adjustments to solar radiation | solar radiation | -10.0—110.0 |
| jh_coef | Coefficient used in Jensen-Haise potential ET computations | Potential ET | 0.005—0.06 |
| jh_coef_hru | Coefficient used in Jensen-Haise potential ET computations | Potential ET | 5.0—25.0 |
| srain_intcp | Summer rain interception storage capacity | interception | 0.0—1.0 |
| wrain_intcp | Winter rain interception storage capacity | interception | 0.0—1.0 |
| cecn_coef | Convection condensation energy coefficient | snow | 2.0—10.0 |
| emis_noppt | Average emissivity of air on days without precipitation | snow | 0.757—1.0 |
| freeh2o_cap | Free-water holding capacity of snowpack | snow | 0.01—0.2 |
| potet_sublim | Snow sublimation fraction of potential ET | snow | 0.1—0.75 |
| carea_max | Maximum area contributing to surface runoff | surface runoff | 0.0—1.0 |
| smidx_coef | Non-linear contributing area coefficient | surface runoff | 0.001—0.06 |
| smidx_exp | Exponent in non-linear contributing area coefficient | surface runoff | 0.1—0.5 |
| fastcoef_lin | Linear coefficient in equation to route preferential-flow | soil-zone | 0.001—0.8 |
| fastcoef_sq | Non-linear coefficient in equation to route preferential-flow | soil-zone | 0.001—1.0 |
| pref_flow_den | Fraction of the soil zone in which preferential flow occurs | soil-zone | 0.0—0.1 |
| sat_threshold | Water capacity between field capacity and total saturation | soil-zone | 1.0—999.0 |
| slowcoef_lin | Linear coefficient for interflow routing | soil-zone | 0.001—0.5 |
| slowcoef_sq | Non-linear coefficient for interflow routing | soil-zone | 0.001—1.0 |
| soil2gw_max | Maximum soil water excess that is routed directly to groundwater | soil-zone | 0.0—0.5 |
| soil_moist_max | Maximum available water holding capacity of soil-zone | soil-zone | 0.001—10.0 |

| | | | |
|---|---|---|---|
| soil_rechr_max | Maximum available water holding capacity of recharge zone | soil-zone | 0.001—5.0 |
| ssr2gw_exp | Non-linear coefficient in equation used to route soil-zone water to groundwater | soil-zone | 0.0—3.0 |
| ssr2gw_rate | Linear coefficient in equation used to route soil-zone water to groundwater | soil-zone | 0.05—0.8 |
| transp_tmax | Temperature that determines start of the transpiration period | soil-zone | 0.0—1000.0 |
| gwflow_coef | Linear groundwater discharge coefficient | groundwater | 0.001—0.5 |

Table 2. Ordered list of most sensitive Precipitation-Runoff Modeling System calibration parameters by process and performance statistic. The parameters listed in each cell of the table are those that are required to account for 90 percent of the cumulative sensitivity across all hydrologic response units (HRUs). The number in parentheses following the parameter name is the proportion of the CONUS HRUs, in percent, in which that parameter is part of the set that accounts for 90 percent of the cumulated sensitivity on an HRU-by-HRU basis. These parameters are described in Table 1.

| Process | Performance Statistic | | |
| --- | --- | --- | --- |
| | Mean | CV | AR 1 |
| Snowmelt | tmax_allsnow(96), tmax_allrain(92) | tmax_allsnow(39), tmax_allrain(38), rad_trncf(9), freeh2o_cap(8), dday_intcp(7) | tmax_allsnow(34), dday_intcp(29), rad_trncf(28), radmax(24), tmax_allrain(17), jh_coef(15), freeh2o_cap(14), cecn_coef(14), emis_noppt(13), jh_coef_hru(13), potet_sublim(10) |
| Surface runoff | smidx_exp(98), carea_max(98), soil_moist_max(98), smidx_coef(96), jh_coef(90), dday_intcp(33) | carea_max(93), smidx_exp(82), jh_coef(64), tmax_allsnow(55), smidx_coef(52), srain_intcp(33), soil_moist_max(23), tmax_allrain(22) | soil_moist_max(92), carea_max(83), jh_coef(65), smidx_exp(64), smidx_coef(42), tmax_allsnow(39), dday_intcp(25), srain_intcp(23), tmax_allrain(16), radmax(15) |
| Infiltration | smidx_exp(99), soil_moist_max(99), carea_max(99), smidx_coef(95), jh_coef(64), srain_intcp(50) | carea_max(80), tmax_allsnow(69), jh_coef(63), smidx_exp(62), srain_intcp(54), smidx_coef(54), tmax_allrain(48), radmax(37), freeh2o_cap(36), soil_moist_max(35), dday_intcp(31), rad_trncf(18) | carea_max(72), soil_moist_max(64), smidx_exp(61), tmax_allsnow(60), srain_intcp(60), tmax_allrain(42), jh_coef(35), smidx_coef(24), freeh2o_cap(16), dday_intcp(16) |
| Soil moisture | soil_moist_max(100), jh_coef(99), | jh_coef(98), radmax(98), soil_moist_max(97), | soil_moist_max(99), jh_coef(98), |

| | | | |
|---|---|---|---|
| | dday_intcp(94), radmax(82) | dday_intcp(94) | dday_intcp(89), radmax(35) |
| Evapo-transpiration | jh_coef (100), soil_moist_max (96), dday_intcp (96), radmax (92), jh_coef_hru (62), smidx_coef (37), dday_slope (25) | radmax(100), jh_coef (100), soil_moist_max (95), dday_intcp (73), dday_slope (67), soil_rechr_max (34) | jh_coef(100), radmax(100), dday_slope(75), soil_moist_max(74), dday_intcp(67), soil_rechr_max(49) |
| Interflow | soil_moist_max(99), soil2gw_max(94), pref_flow_den(90), jh_coef(84), carea_max(65), smidx_exp(45), dday_intcp(31), smidx_coef(19) | fastcoef_lin(100), soil_moist_max(87), pref_flow_den(71), jh_coef(61), carea_max(49), soil2gw_max(29), smidx_exp(25), tmax_allsnow(17), dday_intcp(16) | soil_moist_max(96), fastcoef_lin(89), slowcoef_sq(83), carea_max(72), jh_coef(61), pref_flow_den(47), smidx_exp(47), ssr2gw_exp(40), soil2gw_max(26), dday_intcp(18), tmax_allsnow(16) |
| Baseflow | jh_coef (100), soil_moist_max (91), dday_intcp (81), soil2gw_max (74), radmax (64), carea_max (37, jh_coef_hru (36) | gwflow_coef (48), soil_moist_max (40), jh_coef (28), soil2gw_max (28), smidx_coef(20), carea_max(16), tmax_allsnow(13), dday_intcp(12), smidx_exp (8) | gwflow_coef (48), soil_moist_max (44), soil2gw_max (22), carea_max (18) |
| Runoff | jh_coef(100), dday_intcp(96), soil_moist_max(96), radmax(93), jh_coef_hru(62), smidx_coef(37), dday_slope(26) | gwflow_coef(97), soil_moist_max(81), fastcoef_lin(76), pref_flow_den(71), carea_max(58), jh_coef(54), smidx_exp(49), smidx_coef(42), soil2gw_max(36), tmax_allsnow(15) | slowcoef_sq(90), soil2gw_max(90), gwflow_coef(82), carea_max(81), soil_moist_max(78), smidx_exp(72), smidx_coef(60), fastcoef_lin(36), pref_flow_den(35), jh_coef(30), slowcoef_lin(22) |
| Parameters not sensitive | adjmix_rain, fastcoef_sq, ppt_rad_adj, radj_sppt, radj_wppt, sat_threshold, ssr2gw_rate, tmax_index, transp_tmax, wrain_intcp | | |

1 **Figures**

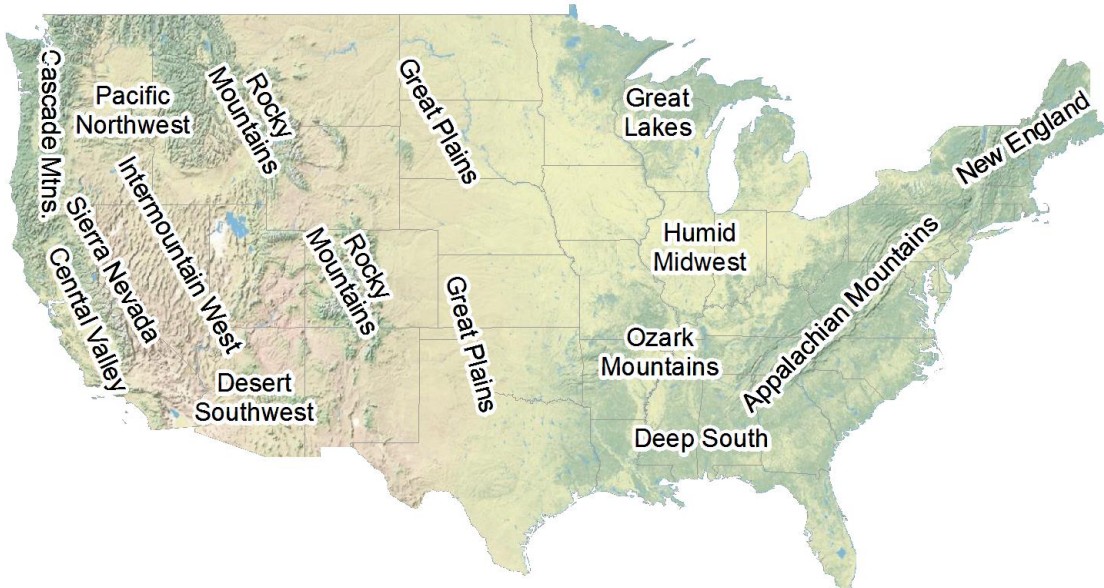

4 Figure 1. Location Map of the conterminous United States showing the different geographic

5 regions referred to this study.

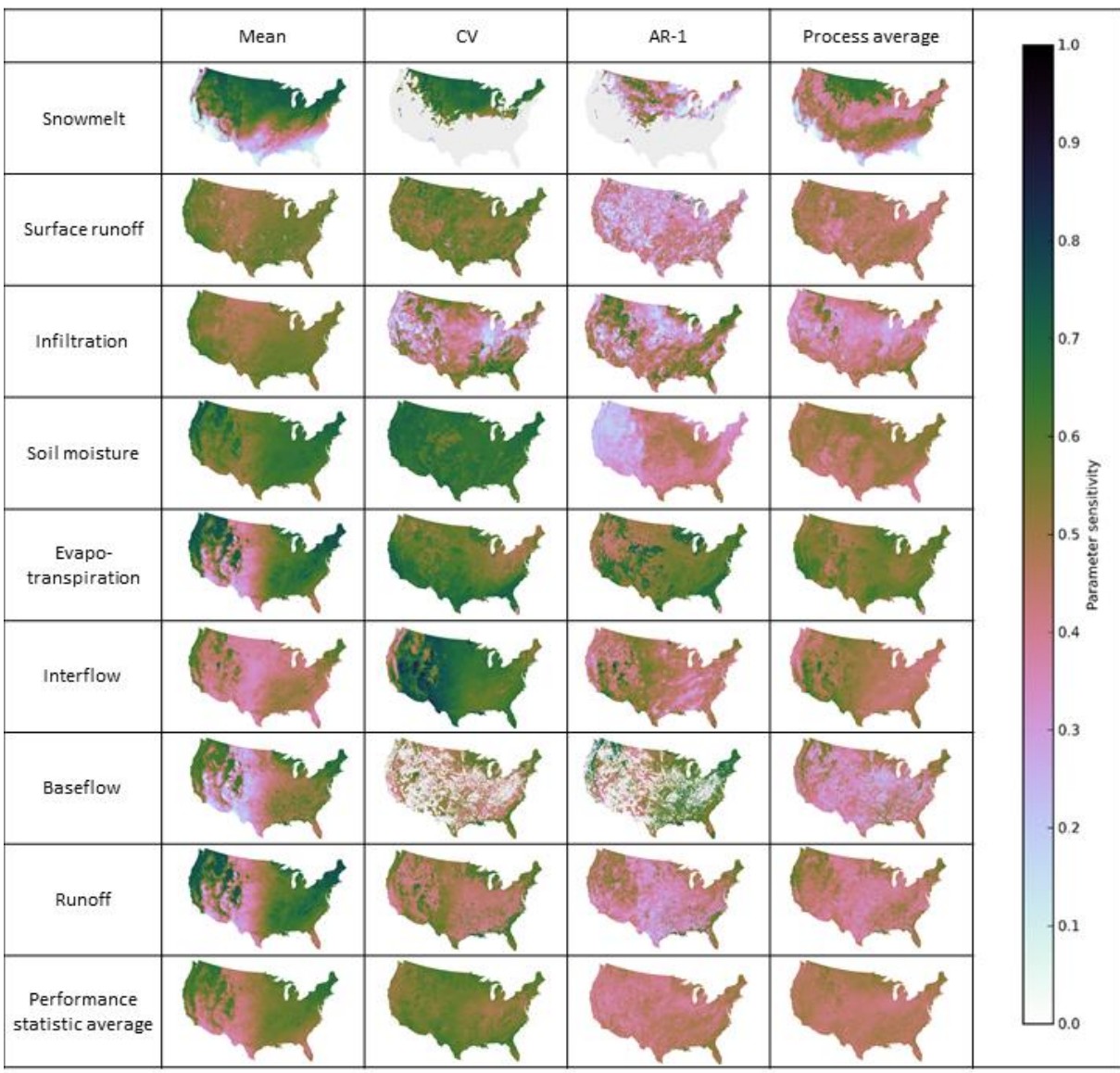

Figure 2. Maps of the conterminous United States showing Precipitation-Runoff Modeling System parameter sensitivity by Hydrologic Response Unit (HRU) by process and performance statistic. The HRUs parameter sensitivity is computed by summing the first-order sensitivity for all parameters. The process average maps are made by averaging the parameter sensitivity values computed for the different performance statistics. The performance statistic maps are made averaging the parameter sensitivity values computed for the different processes.

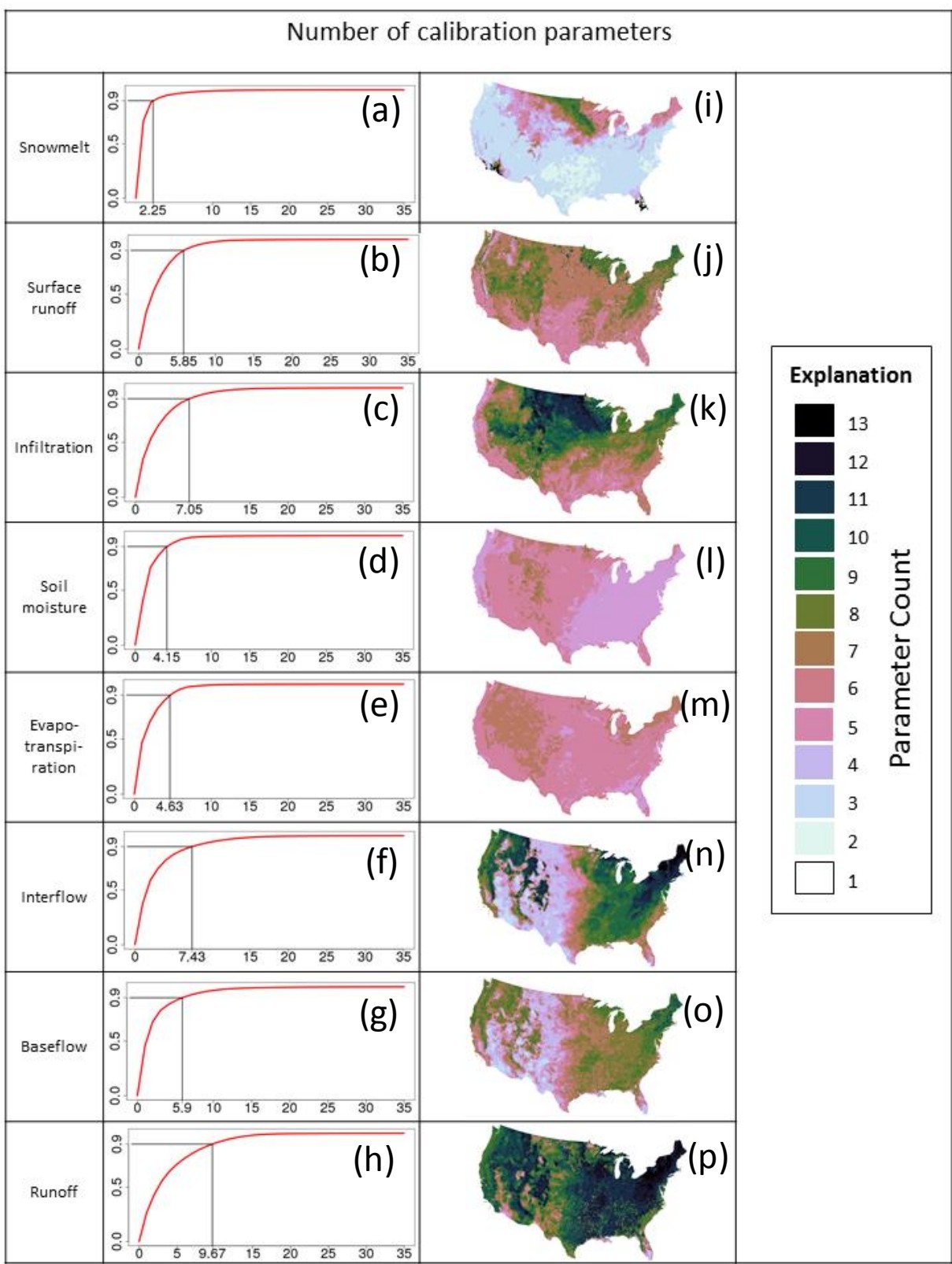

2 Figure 3. Cumulative parameter sensitivity across all Hydrologic Response Units (HRUs) in

3 the CONUS Precipitation-Runoff Modeling System application are shown by process. The

4 plots (a)—(h) show the parameter count necessary to account for 90% of the cumulative

1  parameter sensitivity, summarized across all HRUs. For this count, the parameters are ranked

2  and summed until the 90% level is reached.  The maps (i)—(p) show the count of ranked

3  parameters required to reach the 90% level on an HRU-by-HRU basis, by process.

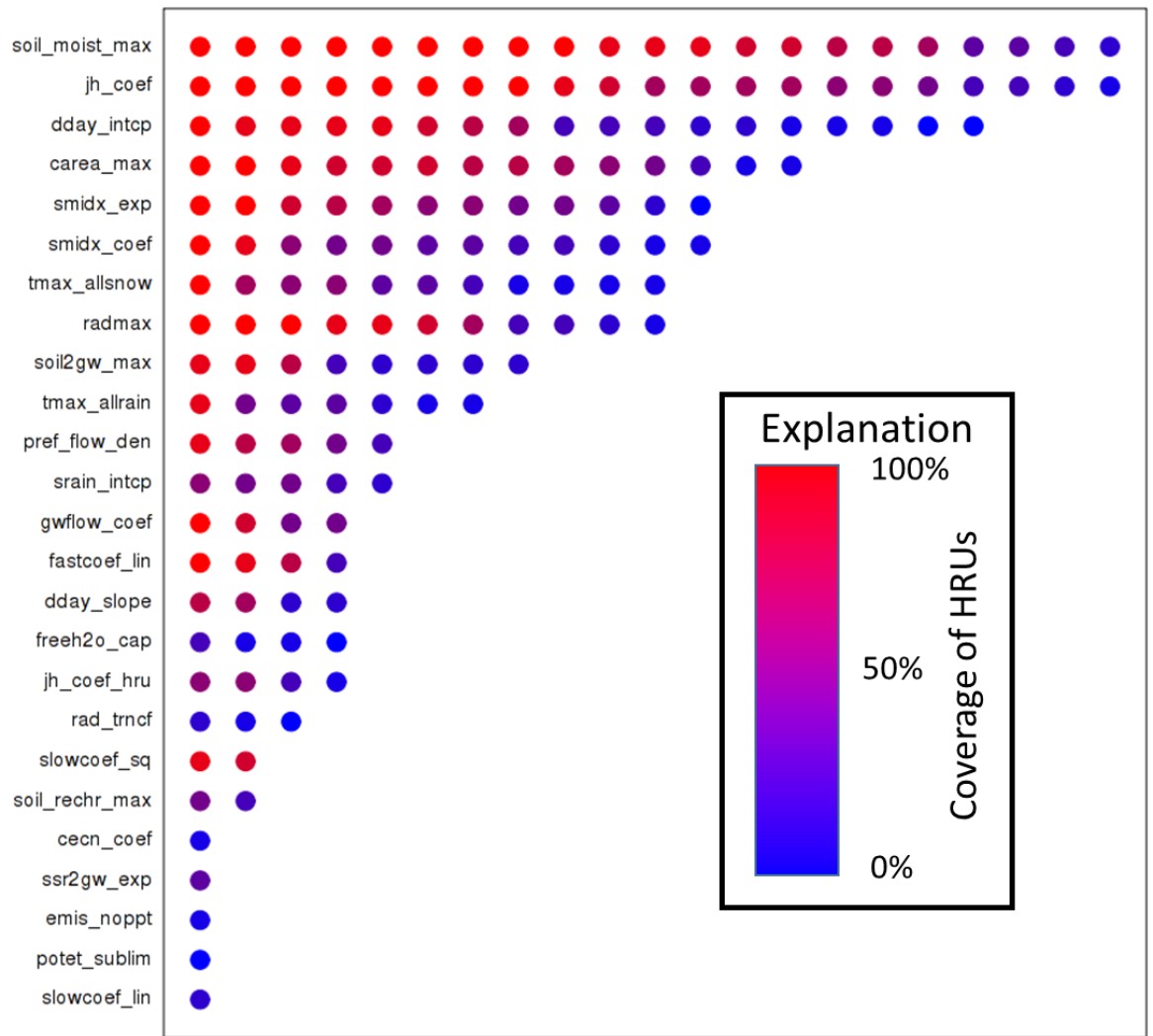

Parameter Occurrence

Figure 4. Summarizes the frequency of occurrence of the different calibration parameters in the process/performance statistic categories of Table 2. The circles in each row adjacent to a parameter name indicate how many times the respective parameter occurs in these different categories. Parameters with more circles are affecting more process categories. The color of each circle indicates the extent of the spatial coverage of that occurrence, specifically, red circles (as opposed to blue) indicate that more Hydrologic Response Units are affected by the respective parameter.

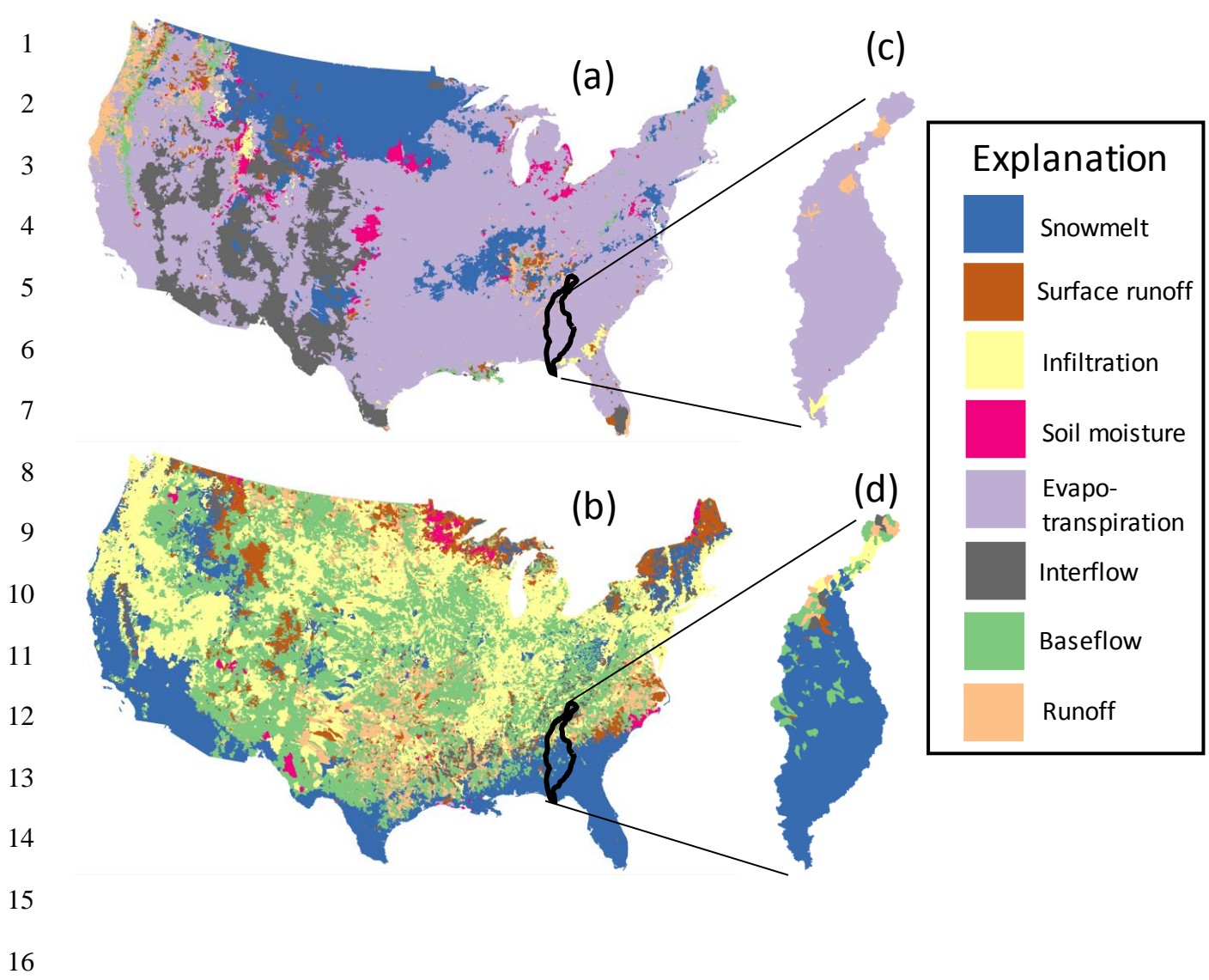

Figure 5. Precipitation-Runoff Modeling System parameter sensitivity organized by process ranked for each hydrologic response unit for the entire conterminous United States (maps (a) and (b)) and for the Apalachicola – Chattahoochee – Flint River basin (maps (c) and (d)). The maps on the top ((a) and (c)) show the most dominant process, while the maps on the bottom ((b) and (d)) show the most inferior process.