# Peer review of "S. L. Markstrom1, L. E. Hay1 and M. P. Clark2"

_Hydrology and Earth System Sciences, 2015_

## Referee Comment (RC1) · S. Höllering (Referee) · 18 Feb 2016

**General comments**

The authors presented an interesting idea of a methodological framework wherein parameters of the HRU based Precipitation-Runoff Modeling System (PRMS) can be identified as influencial in terms of essential hydrological model based processes and statistical streamflow indices serving as objective functions. Parameter influence on model output was evaluated by parameter senstivity index values originating from global sensitivity analysis with the Fourier Amplitude Sensitivity Test (FAST). The approach aims at reducing the number of model input parameters to focus on conceptualised processes assumed as hydrologically relevant within the watersheds of the conterminous United States.

[Figure]

I generally agree with the concept of referencing model response functioning in form of derived objective functions with dependent partial parameter sensitivities for region specific model parameter identification. This is one of the aspects which would be really worth publishing.

Apart from that, fundamental assumptions underlying this study are not sufficiently clarified to address the discussed issues effectively, which are certainly topical and relevant for model based catchment hydrology. The paper is technically well-structured, exhibiting findings of the presented concept concisely but it lacks the required presentation quality at too many different points. However, I found some serious shortcomings and recommend to revise a number of major and minor specific and technical points before the manuscript can be reconsidered for publication.

**Specific comments**

What is the main purpose of your paper?

You mention a number of issues e.g. "parameter identification", "process identification", "calibration advise for modelers" or "identification of [model] structural inadequacies". A better focus on one or two of these issues, preferably on the first and second is advisable here. As uncertainty analysis is not the issue here, I furthermore suggest to remove the part starting from P16L29, which is also rather speculative.

Please also name your assumptions more precisely!

The fundamental assumption of this study is, that the conceptualisation of PRMS is structurally adequate to reproduce all hydrological processes of the CONUS. It is however not adressed, whether this assumption is valid or not or if the study doesn't

claim to be transferable to real world processes and consequently stays a pure virtual PRMS experiment. Conclusions on the dominant hydrological processes are only valid if it is shown that PRMS actually is a good representation of hydrological processes. Processes in the study purely originate from and are defined by the PRMS structure whereby a comparison with observational data might be helpful in this application to show potential deficiencies or justify the fundamental assumption.

P2L19/P10L20: As you similarly found out, more complex processes such as the reproduction of streamflow and its components as well as mountainous regions require more calibration parameters. The general rather small remaining subset of sensitive parameters explaining the majority of the model output variance of processes might be predefined by the conceptual structure of PRMS and a hint to overparameterization. The number of parameters required in a process is also predetermined by the model/process concept and its complexity. Maybe be a bit more specific and less general or sketchy in stating your findings i.e. in the sense of the influence a parameter exerts on a process which might not be purely predetermined by the concept of a model.

P3L13: (How) do these two aspects of complexity correspond to the ones stated in the abstract and explained directly above these lines? Maybe you should be more precise here!

P3L32: This issue has also been partly discussed e.g. by Reusser and Zehe (2011).

P5L8: HRUs are purely derived and defined by their geographic and topographic location. Process identification and catchment classification might be hampered by this definition e.g. by mingling of processes leading to a complex interplay and location specific response behaviour which cannot be always captured by one HRU. In

addition to your discussed points a redefinition of HRUs based on dominant hydrologic processes instead of the applied discretisation based on geographic position might be a conceivable outcome and a consequence of your study maybe helpful for calibration.

P5L20: Here a more precise explanation might be helpful. Is simulated streamflow at locations with stream gauges evaluated differently from streamflow at sites without observations?

P7L1: Here more attention to further studies with streamflow indices could be given (see e.g. Yadav et al. (2007)). Please discuss your choice in some more details.

P9L25: I suggest to start this chapter with the sentence "To identify the expected count of parameters ... (P9L28)" first the theory, then a specific example.

P10L23/P13L8: This view might be kind of model structure/concept specific (as stated above) and is not surprising as streamflow is a convolution of these individual processes. Isn't total HRU runoff in PRMS the pure product or sum of the other streamflow processes (surface runoff, interflow and baseflow), hence involved process parameters add up to a larger number suggesting more complexity? Maybe you can be a bit more precise in the explanations (P13L13).

P15L25: To my knowledge PRMS offers different modules for PET calculations. (How) do sensitivity results and parameter identificaton change by replacing one module by another? This might be subject of future studies and worth mentioning.

P16L3: Someone who is interested in modelling the selected catchment is probably better advised to have a look at historical meteorological observations. From these it should be obvious that snowmelt might not be of any interest here.

**Technical corrections**

*Typing errors:*
The spelling and writing needs improvement and proofreading. To mention several of them:

Please be consistent in the writing and consider HESS manuscript preparation guidelines for authors e.g. Figure, Fig.
P2L15: indicate instead of indicates
P4L3/P16L14: watershed**s**
P8L15: Here poor comprehensibility can be better avoided by changing three to seven objective functions: "... 56 combinations of three objective functions and eight processes (plus totals)."
P11L7: "...**is** surprising..."
P15L12: "This is probably because it **is** a major component of the hydrologic cycle that is..."
P15L21: tha**n**
P16L7: use**d**
P16L11: "...**is** defined..."
P16L14: process**es**

*Reference/citation errors:*
Citations in the manuscript are correct while the year 2014 in complete reference is not:
Markstrom, S. L., Regan, R. S., Hay, L. E., Viger, R. J., Webb, R. M. T., Payn, R. A., and LaFontaine, J. H.: PRMS-IV, the precipitation-runoff modeling system, version 4, U.S. Geological Survey Techniques and Methods, book 6, chap. B7, 158, http://dx.doi.org/10.3133/tm6B7, **2015**

*Figures:*

General remarks:

Resolution and quality of the presented figures and maps seem to be generally not high enough or pixelated and need substantial improvement. Unfortunately, the labeling of latitudinal and longitudinal lines are not readable at all. Please improve the legibility or remove it or incorporate it in only one figure which might be enough to show it once. Some of the shortcomings are listed here:

Figure 1: This map lacks both sufficient quality and a valuable information content. In my oppinion a different form of presentation such as histograms or kernel density estimates for selected attributes of HRUs could be beneficial.

Figure 2: Please use consistent spelling or abbreveations for objective functions across tables and figures. Please explain the additional column "Process average" in the results section 4.1 and the meaning of the legend.

The caption should also provide more information.

Figure 3: Better use as Figure 1. It furthermore contains little information and poor legibility of region names.

Figure 4: "The plots A-**H** summarize..."

Figure 5: Please clarify the connection to the ordered listing of Table 1.

Figure 6: Please raise font sizes of titles above each map to be readable or remove them from the figure.

**References**

Reusser, D. E. and Zehe, E.: Inferring model structural deficits by analyzing temporal dynamics of model performance and parameter sensitivity, Water Resources Research, 47, doi:10.1029/2010WR009946, 2011.

Yadav, M., Wagener, T., and Gupta, H.: Regionalization of constraints on expected watershed response behavior for improved predictions in ungauged basins,

Advances in Water Resources, 30, 1756–1774, doi:10.1016/j.advwatres.2007.01.005, 2007.

---

## Referee Comment (RC2) · B. Guse (Referee) · 23 Feb 2016

This article applies a parameter sensitivity analysis using the FAST algorithm on different model outputs of a distributed-parameter hydrology model (DPHM). It focuses on presenting spatial patterns of parameter sensitivities for the different model outputs and extracts the dominant processes for United States of America.

I really like the core idea of this study and think this manuscript is worth to be published. However, in my opinion, the article might benefit from a couple of improvements to emphasize the major outcomes more precisely. I also think that the interpretation and discussion of the results could be more clearly to make the study interesting for a broader audience. At some points, I presented some ideas which might worth discussing about it. Thus, I encourage the authors to consider the following remarks.

[Figure]

Major comments:

I encourage the authors to improve the readibility of the abstract to present the idea of this study in a clearer way.

Please think about the use of the notation "objective function" for mean, CV,... . In my understanding, these are statistical values describing different model outputs without giving information of the model performance. The use of the term "objective function" indicates an evaluation of the model performance according its common use in hydrological modelling. I propose to use "fundamental daily streamflow statistics (FDSS)" as mentioned in the text instead of "objective function".

A table with the model parameters and their corresponding processes is missing. I see that you refer to another article. However, this manuscript would be more readable, if the reader has an idea of the parameter used for this study. When stating that a certain number of parameters is required "to account for 90% of the parameter sensitivity" is necessary to know how many parameters for this process are included in the model structure. For example, assuming that there are only two snow parameters, then it is not surprising when the number of required parameters is two. However, let's say that are eight parameters for the snow process then it is interesting to know that only two parameters are required.

Furthermore, in chapter 4.2, you should mention whether the parameters (accounting for 90%) are identical for a certain process or vary (P. 10, L.5-6).

It is really interesting to see a systematic in the number of parameters as stated on P. 10, L.20-23. Could you explain it? At best in relation to the model structure? Are you expect a different result for different models (structures)?. While this result is reasonable for snowmelt, it is really surprising that you only need a small number of parameters to explain the soil moisture behaviour.

I think that the article would benefit if you could relate the results (e.g. P.10, L.24-30)
to the process heterogenity in the different parts of the CONUS. There are certainly regions with very complex process patterns and other with a clear dominance of a single process. Are there other studies looking at process dominance or process heterogeneity in the CONUS? Maybe you can make a comparison with these studies?

It is certainly required to discuss the relationship of model parameters and the corresponding processes. The stronger this relationship is, the more sensitive a parameter might be for this process. Could you mention how the parameter-process relationship affect your results?

By summing up the first-order partial variance and using this value as indicator to estimate the dominant process, you do not consider the parameter interactions (second and higher order sensitivities). However, the parameter interaction depends (among others) on the parameter selection. Could you explain how this aspect affect you results?

The interpretation of table 1 needs to be reworked. I do not agree at least with the sentence on P. 11, L.16-18 that a count of dominant parameters shows how important a parameter is. Assuming that a parameter is strongly related to a certain process, e.g. snowmelt, and is thus relevant for the three objective functions related to snowmelt, but not to the other processes (maybe except of runoff), it is still an important parameter for this specific process. This interpretation and also of the fig. 5 aggregates the results in my opinion in a strong way. It might be more interesting to look at the relationship of model parameters to the processes. To how many processes you can related a parameter? Are these results reasonable when looking at the model structure? An idea of how to relate model parameters and corresponding processes is given in the figures and tables in Pfannerstill et al. (2015).

Concerning the discussion of the spatial heterogeneity in parameter sensitivity (subchapter 5.1), it might worth looking at the expert knowledge on dominant processes in the CONUS. It is not surprising when a HRU with a complex hydrological situation with

relevant contributions from different runoff components provides a different results as a HRU with a strong dominance of one hydrological component. Here, I think that a general discussion of process dominance is missing and a discussion in the context of former studies on dominant processes in the CONUS (if existing).

Maybe you can think about presenting the results in Tab. 1 and Figs. 4 and 5 in a different way, so that the most important outputs are more emphasized. It is rather difficult to extract information of the relationship of parameter and processes from Tab. 1 and a counting how often a parameter occurs is also time-consuming. But in my opinion this information is required to make Fig. 5 more informative.

Fig. 4: Is it maybe relevant thinking about the variability, e.g. in the snowmelt subplot? It is stated that on average 2.25 parameters are required to explain 90%. The map (subplot 4M) shows that in most of the HRUs only 2 or 3 parameters are required. However in the snow-dominated northern parts up to 10 parameters are required. It might be worth thinking about extracting additional information from this idea. One way would be to add an additional line in the subplots 4A-4H which is only related to HRUs which have certain relevance of this process (kind of threshold exceedance approach or something similar).

Fig. 6: Could you explain why infiltration is the inferior process in many HRUs. I cannot imagine a hydrological situation in which the infiltration process is less relevant than total runoff, all runoff components, ETP, soil moisture.

It might be interesting to think about the following results of the Fig 4-5: According to Fig. 4 only 4.15 parameters are required to explain soil moisture, which is a relative low value keeping in mind that the soil moisture interacts with almost all other processes. Furthermore, there are 7.05 parameters needed for infiltration. Then, it is stated in Fig. 5 that soil_moist_max is overall the most important parameter. Do this mean that the relationship between soil_moist_max and soil moisture is extremely high so that only a few additional parameters (about 3) are needed to reproduce the soil moisture

conditions?

Minor comments:

Abstract:

Page 2, Line 2: The first sentence of the abstract could be written more clearly. Why not only writing: "The Precipitation-Runoff Modeling System as a distributed-parameter hydrologic model has been applied to the conterminous United States.

P. 2, L. 4-5: Whilst it is certainly clear that the number of parameters is an aspect of model complexity, this is not fully clear for the "interpretation of the model output". Is this really an aspect of complexity? Do you assume that the model which provides a higher number of model outputs is more complex?

P. 2, L. 5-8: To make the abstract more readable, I would suggest to subdivide this sentence into two separate ones. There are too many aspects in this sentence (parameter sensitivity for simplification, parameter identification and its relationship to dominant processes, spatial patterns)

P. 2, L. 9-10: I do not think that this sentence is understandable when reading the abstract at first before knowning the whole article. What do you mean with "processes correspond to variables"? Which type of variables?

P. 2, L. 11: The notation "categories" is not clearly described in the abstract.

P. 2, L. 12-13: How do you estimate the "model performance" by visualizing categories? This part needs to be improved.

P. 2, L. 16: The benefit of a reduction of the dimensionality of output variables or objective functions is not clear.

P. 2, L. 22: I would encourage the authors to add a final sentence to emphasise the general advantage of this study.

Introduction:

P. 2, L. 28: The article would be benefit from a clear definition of "input parameters". Is an input parameter related to a driver of the hydrologic cycle such as precipitation or solar radiation or more to a real model parameter? In all cases, it is better to avoid potential misunderstandings.

P. 3, L. 1: References are missing such as for constraining parameter in models, e.g. Hrachowitz et al. (2014) and for stating that different parameter good have a comparable impacts on the model results.

P. 3, L. 6: The three references are related to studies which investigate performance measures more precisely. It might be good to also have a reference to studies which are directly investigating the model output.

P. 3, L. 11-12: Please also add the study from Reusser et al. (2009).

P. 3, L. 14: Please indicate that you consider uncertainty in this study only on input parameter uncertainty and not on structural uncertainty in the model.

P. 3, L. 18-28: It might be good to mention here that it is at least at this scale impossible to support the results with adequate measurements in addition to the total discharge.

P. 4, L. 1: References are here missing, e.g. Wagener et al. (2003), Reusser et al. (2011), Guse et al. (2014).

P. 4, L. 11: Reference of Reusser et al. (2011) is missing.

P. 4, L. 20-22: As mentioned before, it is not clear why you aimed "to reduce the number of inputs and outputs". I think the overall aim should be a clearer characterization of the model parameters and to focus on the dominant processes.

Methods:

P. 4, L.29- P. 6, L.7: Please check carefully if you could reduce the subchapter 2.1 in

length. Do you really need this information for this article?

P. 6, L.8-25: The selection of the eight output variables is reasonable and seems to be representative for hydrological studies with distributed models. Maybe you can emphasize this to give the article a more general character.

P. 7., L. 18: Please also add the reference of Guse et al., 2014, since it is the initial study for Pfannerstill et al. 2015.

Results:

P. 8, L. 17: Please think about a more precise title for the subchapter 4.1.

P. 8, L. 20-23: This sentence is not understandable. It is understandable that you have calculated the sum of the first-order partial variance. However, it is not clear how you can estimate an average value (average of what?).

P. 8, L. 23: The total sensitivity is one, is it? Why do you need to scale the sum of the sensitivities to the total sensitivity?

P. 8, L. 23: "category of modeled process" instead of "category of process".

P. 8, L.28-30: I recommend to be more precisely here: You have calculated the sum of all partial sensitivities for a certain HRU for each process. Then, the process with the highest sum of the first-order sensitivity is indicated as "dominant process". To make this clear, you should add that the dominant process is the process with the largest sum of all first-order partial variances (sensitivities). This is required since the sensitivity of a single parameter is not shown here.

P. 9, L.17-18: Can you extract a systematic pattern in these results?

P. 10, L.24-25: Please add that this statement is not valid (or only to a low extent) to fig 4J and 4N.

P. 11, L. 6-9: Do you see a general systematic why the spatial patterns of parameter

sensitivity are different for the different objective functions. It might be interesting to give further statements on this.

P.11, L. 28-32: When stating that the parameter "soil_moist_max" is the most important and a model calibration should be focused on it, then it is required to know for which process this parameter is relevant. Assuming that a typical calibration uses discharge as target variable, a focus on "soil_moist_max" helpful in the case of a dominance of "soil_moist_max" on runoff. However, to include this information in a calibration in the case of a dominance on other process but not on runoff?

P. 12, L.2-8: The part on the least sensitive parameter can be removed since the reader does not receive any details about the parameters. Or could you extract some further information from the fact that these parameters have a low sensitivity?

P. 12, L. 9-14: I think that the authors should add here some more details. It is really helpful if a parameter can be precisely characterized by saying that it is only dominant in a very specific case (e.g. for one process). But this information cannot currently not be extracted from article.

P. 13, L.8-12: I like this part. Maybe you can in addition relate it to the concept of vertical water redistribution (Yilmaz et al., 2008, Pfannerstill et al., 2015).

P. 14, L. 22-23, Step 1: Summed in time?

P. 14, L. 24-25, Step 2: How to you obtain a score for each process? Do you assign each parameter to a certain process? If yes, then you have to mention somewhere which parameter is related to which process.

P. 16, L. 31: Spelling error: Mishra (2009)

Figures:

Fig. 1: Could be removed. I do not see an advantage of it. Maybe you can transfer it to the supplementary material.

Fig. 2: Does the last row and column present the average values along the row/column? Do you maybe have to change "process average" and "objective function average"?

I recommend to show the figure 3 before the figure 2, since fig. 3 provide a general map of the USA whilst, fig. 2 already show the distributed results.

Figure 4 would benefit from knowning which parameters are within the 90% and how variable the parameters belonging to this 90% are?

Fig. 4: The legend needs to be graphically improved.

I do not really see a real benefit of fig. 5. Maybe you can extract the results in a better way. One point might be that the model parameters are not explained and even the related processes are not highlighted in Fig. 5. In particular, it is not clear which information you can derive from the last place occurrence.

It is not fully clear which information you can derived from investigating the most inferior process. It seems to be that this is either clear such as snowmelt parameter for California or related to the model structure.

Reference list:

Guse, B., Reusser, D. E., and Fohrer, N.: How to improve the representation of hydrological processes in SWAT for a lowland catchment – Temporal analysis of parameter sensitivity and model performance, Hydrol. Process., 28, 2651–2670, doi:10.1002/hyp.9777, 2014.

Hrachowitz, M., O. Fovet, L. Ruiz, T. Euser, S. Gharari, R. Nijzink, J. Freer, H. H. G. Savenije, and C. Gascuel-Odoux: Process consistency in models: The importance of system signatures, expert knowledge, and process complexity, Water Resour. Res., 50, doi:10.1002/2014WR015484, 2014

Pfannerstill, M., Guse, B., Reusser, D., and Fohrer, N.: Process verification of a hydrological model using a temporal parameter sensitivity analysis. Hydrology and Earth System Sciences 19: 4365–4376, 2015.

Reusser, D. E., Blume, T., Schaefli, B., and Zehe, E.: Analysing the temporal dynamics of model performance for hydrological models, Hydrol. Earth Syst. Sci., 13, 999–1018, doi:10.5194/hess-13-999-2009, 2009.

Reusser, D.E., and Zehe, E.: Inferring model structural deficits by analyzing temporal dynamics of model performance and parameter sensitivity. Water Resources Research 47(7): W07550. DOI:10.1029/2010WR009946, 2011.

Wagener, T., McIntyre, N., Lees, M.J., Wheater, H.S., Gupta, H.V.: Towards reduced uncertainty in conceptual rainfall–runoff modelling: dynamic identifiability analysis. Hydrological Processes 17: 455–476, 2003.

Yilmaz, K. K., Gupta, H. V., and Wagener, T.: A process-based diagnostic approach to model evaluation: Application to the NWS distributed hydrologic model, Water Resour. Res., 44, W09417, doi:10.1029/2007WR006716, 2008.

---

## Author Comment (AC2) · 21 Jun 2016

The comment was uploaded in the form of a supplement:
http://www.hydrol-earth-syst-sci-discuss.net/hess-2015-508/hess-2015-508-AC2-supplement.pdf

---

## Author Response (AR1)

**Black text: B. Guse's comments**

**Red text: S. Markstrom's response**

I added numbers to the comments so I could more easily refer to them.

I apologize for misspelling Dr. Guse's name in a previous version of this response to his comments.

Major comments:

I encourage the authors to improve the readibility of the abstract to present the idea of this study in a clearer way.

Yes, I accept your specific comments below related to the abstract. In addition, I have rewritten much of the text.

1. Please think about the use of the notation "objective function" for mean, CV,... . In my understanding, these are statistical values describing different model outputs without giving information of the model performance. The use of the term "objective function"

indicates an evaluation of the model performance according its common use in hydrological modelling. I propose to use "fundamental daily streamflow statistics (FDSS)" as mentioned in the text instead of "objective function".

Accepted. Yes, I agree that confusion may arise from the non-standard use of "objective function." I changed to the term "performance measure" as I want to emphasize that this is really a measure of how the model preforms in relation to the parameter values. Also, "performance measure" seems to make the text flow better.

2. A table with the model parameters and their corresponding processes is missing. I see that you refer to another article. However, this manuscript would be more readable, if the reader has an idea of the parameter used for this study. When stating that a certain number of parameters is required "to account for 90% of the parameter sensitivity" is necessary to know how many parameters for this process are included in the model structure. For example, assuming that there are only two snow parameters, then it is not surprising when the number of required parameters is two. However, let's say that are eight parameters for the snow process then it is interesting to know that only two parameters are required.

Yes, I added a table (table 1) that lists all of the calibration parameters, description, and what I called "PRMS module type". This PRMS module type is what I believe you are asking for in your comment. I did not want to call this "process" because I did not want to confuse the reader with the sensitivity analysis based "process identification" the is performed on the PRMS output.

Now, table 2 does show the parameters and the corresponding identified processes, but this was determined by the sensitivity analysis. Processes identified here make no a priori assumptions about which parameters may affect any particular process. For instance, PRMS uses a potential evapotranspiration coefficient parameter. Clearly, this parameter can be directly associated with the "transpiration process", but to what degree is this parameter associated with the "snowmelt process"? PRMS does simulate snow sublimation, but a priori, should the potential ET coefficient be considered a "snow melt parameter"? Because of the unknown relationships in model structure, this must be determined with the global parameter sensitivity analysis, and that is the point of table 1.

3. Furthermore, in chapter 4.2, you should mention whether the parameters (accounting for 90%) are identical for a certain process or vary (P. 10, L.5-6).

That is the information I try to convey in table 2. The problem is that spatially (on an HRU-by-HRU basis), which specific parameters make up the 90% could vary. Table 2 summarizes this across all HRUs for the CONUS. The idea that I was trying to get across is that the number of parameters needed to characterize a process is some measure of the complexity of that process, and that complexity varies by process and spatially by region of the CONUS. Table 2 summarizes this in a general way so that PRMS modelers could have some idea about which parameters to actually use in their models.

To address this, I added he percent of the CONUS HRUs in which that parameter is part of the set that accounts for 90 percent of the cumulated sensitivity on an HRU-by-HRU basis to the parameter names listed in table 2. I hope this addresses comment 3 by showing which ones vary the most.

4. It is really interesting to see a systematic in the number of parameters as stated on P.

10, L.20-23. Could you explain it? At best in relation to the model structure? Are you expect a different result for different models (structures)?. While this result is reasonable for snowmelt, it is really surprising that you only need a small number of parameters to explain the soil moisture behaviour.

Yes, I added the sentence: "An analysis of theses parameter counts and how they relate to their respective process is beyond the scope of this article, but it could relate to the structure of PRMS and possibly indicate that some processes are over parameterized."

5. I think that the article would benefit if you could relate the results (e.g. P.10, L.24-30)

to the process heterogenity in the different parts of the CONUS. There are certainly regions with very complex process patterns and other with a clear dominance of a single process. Are there other studies looking at process dominance or process heterogeneity in the CONUS? Maybe you can make a comparison with these studies?

I am unaware of studies which classify watersheds (regions, HRUs, etc.) necessarily by process (e.g. "snowmelt watersheds"). Some studies that I am aware of tend to classify space by mappings of soil, geology, vegetation, etc. or properties of driving climate data. These tend to use a principle components type analysis, so there are distinct classifications, but these classifications can not necessarily be related to a dominate process. Other studies tend to be based on streamflow statistics for dendritic grouping. This method seems to be effective for classification, but not necessarily classes that are associated with obviously identifiable processes.

6. It is certainly required to discuss the relationship of model parameters and the corresponding processes. The stronger this relationship is, the more sensitive a parameter might be for this process. Could you mention how the parameter-process relationship affect your results?

Yes, the other reviewer suggested that I focus more on "parameter identification" and "process
identification." I think this is related to your comment here. I rewrote the Introduction with this in mind.

7. By summing up the first-order partial variance and using this value as indicator to estimate the dominant process, you do not consider the parameter interactions (second and higher order sensitivities). However, the parameter interaction depends (among others) on the parameter selection. Could you explain how this aspect affect you results?

Yes, to section 3, I added: "An important caveat is that these higher order variances are not accounted for
in the analysis. It is assumed that first-order partial variance is sufficient to identify sensitive parameters.
This same assumption, as applied to process identification, may be more problematic. If there are sets of
interactive sensitive parameters that have not been identified, then the associated process(es) will not be
identified as such."

8. The interpretation of table 1 needs to be reworked. I do not agree at least with the sentence on P. 11, L.16-18 that a count of dominant parameters shows how important a parameter is. Assuming that a parameter is strongly related to a certain process, e.g.

snowmelt, and is thus relevant for the three objective functions related to snowmelt, but not to the other processes (maybe except of runoff), it is still an important parameter for this specific process. This interpretation and also of the fig. 5 aggregates the results in my opinion in a strong way. It might be more interesting to look at the relationship of model parameters to the processes. To how many processes you can related a parameter? Are these results reasonable when looking at the model structure? An idea of how to relate model parameters and corresponding processes is given in the figures and tables in Pfannerstill et al. (2015).

Yes, I think the problem is my use of the word "important." This is not the right word. I have rewritten these
sentences. Hopefully it is clearer. Figure 5 does show how many processes are identified (related to) a
parameter. I hope that my rewritten description makes this issue clearer.

9. Concerning the discussion of the spatial heterogeneity in parameter sensitivity (subchapter

5.1), it might worth looking at the expert knowledge on dominant processes in the CONUS. It is not surprising when a HRU with a complex hydrological situation with relevant contributions from different runoff components provides a different results as a HRU with a strong dominance of one hydrological component. Here, I think that a general discussion of process dominance is missing and a discussion in the context of former studies on dominant processes in the CONUS (if existing).

See response to comment 5 about other studies.

10. Maybe you can think about presenting the results in Tab. 1 and Figs. 4 and 5 in a different way, so that the most important outputs are more emphasized. It is rather difficult to extract information of the relationship of parameter and processes from Tab. 1 and a counting how often a parameter occurs is also time-consuming. But in my opinion this information is required to make Fig. 5 more informative.

I'm not sure how to do this. The most important outputs, in my opinion, are to give the modeler versions of the table and figures exclusively for the area that he is modeling. And I have been doing this for the people that I work with. For this article, the problem is that I have to keep it general for all of CONUS.

Fig. 4: Is it maybe relevant thinking about the variability, e.g. in the snowmelt subplot? It is stated that on average 2.25 parameters are required to explain 90%. The map (subplot 4M) shows that in most of the HRUs only 2 or 3 parameters are required. However in the snow-dominated northern parts up to 10 parameters are required. It might be worth thinking about extracting additional information from this idea. One way would be to add an additional line in the subplots 4A-4H which is only related to HRUs which have certain relevance of this process (kind of threshold exceedance approach or something similar).

I believe that I have addressed this issue of HRU parameter variability in table 2 and the text I added in relation to figure 5.

11. Fig. 6: Could you explain why infiltration is the inferior process in many HRUs. I cannot imagine a hydrological situation in which the infiltration process is less relevant than total runoff, all runoff components, ETP, soil moisture.

It's not that infiltration is not important, it's just that the sensitivity analysis indicates that there are no parameters that can be changed to affect the model output. Also, there are often multiple processes that are pretty much at the same level of "inferiorness" and one has to be the most. In a very preliminary draft I had version of these maps that showed, for each HRU, the two most inferior process, the three most, etc. These maps really confused my co-authors and in the end, I dropped them.

12. It might be interesting to think about the following results of the Fig 4-5: According to Fig. 4 only 4.15 parameters are required to explain soil moisture, which is a relative low value keeping in mind that the soil moisture interacts with almost all other processes. Furthermore, there are 7.05 parameters needed for infiltration. Then, it is stated in Fig. 5 that soil_moist_max is overall the most important parameter. Do this mean that the relationship between soil_moist_max and soil moisture is extremely high so that only a few additional parameters (about 3) are needed to reproduce the soil moisture conditions?

Yes, I think this interpretation is correct. A source of confusion could be my use of the word "important." In retrospect, that is a loaded word. See my response to your comments number 8 and 11.

Minor comments:

Abstract:

Page 2, Line 2: The first sentence of the abstract could be written more clearly. Why not only writing: "The Precipitation-Runoff Modeling System as a distributed-parameter hydrologic model has been applied to the conterminous United States.

Yes, accepted.

P. 2, L. 4-5: Whilst it is certainly clear that the number of parameters is an aspect of model complexity, this is not fully clear for the "interpretation of the model output". Is this really an aspect of complexity? Do you assume that the model which provides a higher number of model outputs is more complex?

Yes, rewritten. I'm trying to establish the point that by identifying the dominate processes (with respect to PRMS), users can focus on the output variables related to those processes.

P. 2, L. 5-8: To make the abstract more readable, I would suggest to subdivide this sentence into two separate ones. There are too many aspects in this sentence (parameter sensitivity for simplification, parameter identification and its relationship to dominant processes, spatial patterns)

Yes, accepted.

P. 2, L. 9-10: I do not think that this sentence is understandable when reading the abstract at first before knowning the whole article. What do you mean with "processes correspond to variables"? Which type of variables?

Yes, changed this sentence.

P. 2, L. 11: The notation "categories" is not clearly described in the abstract.

Yes, changed.

P. 2, L. 12-13: How do you estimate the "model performance" by visualizing categories? This part needs to be improved.

Yes, changed.

P. 2, L. 16: The benefit of a reduction of the dimensionality of output variables or objective functions is not clear.

Yes. changed.

P. 2, L. 22: I would encourage the authors to add a final sentence to emphasise the general advantage of this study.

Yes, added.

Introduction:
P. 2, L. 28: The article would be benefit from a clear definition of "input parameters".
Is an input parameter related to a driver of the hydrologic cycle such as precipitation or solar radiation or more to a real model parameter? In all cases, it is better to avoid potential misunderstandings.

Yes, added.

P. 3, L. 1: References are missing such as for constraining parameter in models, e.g. Hrachowitz et al. (2014) and for stating that different parameter good have a comparable impacts on the model results.

Yes, added.

P. 3, L. 6: The three references are related to studies which investigate performance measures more precisely. It might be good to also have a reference to studies which are directly investigating the model output.

Yes, added.

P. 3, L. 11-12: Please also add the study from Reusser et al. (2009).

Yes, added.

P. 3, L. 14: Please indicate that you consider uncertainty in this study only on input parameter uncertainty and not on structural uncertainty in the model.

These lines were deleted in response to comments by another reviewer.

P. 3, L. 18-28: It might be good to mention here that it is at least at this scale impossible to support the results with adequate measurements in addition to the total discharge.

These lines were deleted in response to comments by another reviewer.

P. 4, L. 1: References are here missing, e.g. Wagener et al. (2003), Reusser et al. (2011), Guse et al. (2014).

Yes, added.

P. 4, L. 11: Reference of Reusser et al. (2011) is missing.

Yes, added.

P. 4, L. 20-22: As mentioned before, it is not clear why you aimed "to reduce the number of inputs and outputs". I think the overall aim should be a clearer characterization of the model parameters and to focus on the dominant processes.

Yes, I reworded this sentence.

Methods:
    1.  P. 4, L.29- P. 6, L.7: Please check carefully if you could reduce the subchapter 2.1 in length. Do you really need this information for this article?

Yes, this section has been reorganized.

P. 6, L.8-25: The selection of the eight output variables is reasonable and seems to be representative for hydrological studies with distributed models. Maybe you can emphasize this to give the article a more general character.

Yes, added.

P. 7., L. 18: Please also add the reference of Guse et al., 2014, since it is the initial study for Pfannerstill et al. 2015.

Yes, added.

Results:
    1.  P. 8, L. 17: Please think about a more precise title for the subchapter 4.1.

Yes, changed it to "Parameter sensitivity by process and performance measure"

2. P. 8, L. 20-23: This sentence is not understandable. It is understandable that you have calculated the sum of the first-order partial variance. However, it is not clear how you can estimate an average value (average of what?).

Yes, the meaning of the text here is not clear to you. I have added several sentences here to make this clearer.

3. P. 8, L. 23: The total sensitivity is one, is it? Why do you need to scale the sum of the sensitivities to the total sensitivity?

The sum of the individual sensitivities is not necessarily one. If none of the parameters are sensitive than the sum of the parameter sensitivities will be closer to zero.

4. P. 8, L. 23: "category of modeled process"instead of "category of process".

Yes, accepted.

5. P. 8, L.28-30: I recommend to be more precisely here: You have calculated the sum of All partial sensitivities for a certain HRU for each process. Then, the process with the Highest sum of the first-order sensitivity is indicated as "dominant process". To make This clear, you should add that the dominant process is the process with the largest sum Of all first-order partial variances (sensitivities). This is required since the sensitivity of A single parameter is not shown here.

Yes, reworded these sentences.

P. 9, L.17-18: Can you extract a systematic pattern in these results?

Yes, added ", and humid versus arid climates." to the previous sentence.

P. 10, L.24-25: Please add that this statement is not valid (or only to a low extent) to fig 4J and 4N.

Yes, added this.

P. 11, L. 6-9: Do you see a general systematic why the spatial patterns of parameter Sensitivity are different for the different objective functions. It might be interesting to Give further statements on this.

There are certainly patterns here and I very much agree that they are interesting. I have not had time to investigate this properly and would prefer to leave statements about this out of this article rather than speculate.

There is clearly a swath of sensitivity that goes through the Great Plains. Many hydrologic modelers in the US have noted that this area is notoriously difficult to model with physical, statistical, etc. models – and no one is really sure why this is. Our group has a PhD student who is looking into this. Maybe a subsequent article can address this further.

P.11, L. 28-32: When stating that the parameter "soil_moist_max" is the most important and a model calibration should be focused on it, then it is required to know for which process this parameter is relevant. Assuming that a typical calibration uses discharge as target variable, a focus on "soil_moist_max" helpful in the case of a dominance of

"soil_moist_max" on runoff. However, to include this information in a calibration in the case of a dominance on other process but not on runoff?

Yes, I rewrote this paragraph based on comments from the other reviewer. I believe my revision addresses this comment as well.

P. 12, L.2-8: The part on the least sensitive parameter can be removed since the reader does not receive any details about the parameters. Or could you extract some further information from the fact that these parameters have a low sensitivity?

Yes, I now say that modelers should leave them at default values because there is limited information to calibrated them.

P. 12, L. 9-14: I think that the authors should add here some more details. It is really helpful if a parameter can be precisely characterized by saying that it is only dominant in a very specific case (e.g. for one process). But this information cannot currently not be extracted from article.

This varies by HRU/geographic region, so it is difficult to provide specific calibration instructions for the whole of the CONUS. I do provide exactly this type of information on an application site by application site basis to the modelers that I work with. I'm uncertain how to put this information into this article.

P. 13, L.8-12: I like this part. Maybe you can in addition relate it to the concept of vertical water redistribution (Yilmaz et al., 2008, Pfannerstill et al., 2015).

Yes, I added a sentence about this.

P. 14, L. 22-23, Step 1: Summed in time?

Yes, added.

P. 14, L. 24-25, Step 2: How to you obtain a score for each process? Do you assign each parameter to a certain process? If yes, then you have to mention somewhere which parameter is related to which process.

Please see my response to your comments 2 and 3 ("Major" comments section), and 2 and 3 in the "Results" comments section.

P. 16, L. 31: Spelling error: Mishra (2009)

On recommendation of other reviewer, I removed this paragraph.

Figures:
Fig. 1: Could be removed. I do not see an advantage of it. Maybe you can transfer it to the supplementary material.

Yes, removed.

Fig. 2: Does the last row and column present the average values along the row/column? Do you maybe have to change "process average" and "objective function average"?

Please see response to "Major" comments 2 and 3.

I recommend to show the figure 3 before the figure 2, since fig. 3 provide a general map of the USA whilst, fig. 2 already show the distributed results.

Yes, moved figure 3 to figure 1 (after deleting old figure 1).

Figure 4 would benefit from knowing which parameters are within the 90% and how variable the parameters belonging to this 90% are?

Yes, see my response to comment 10.

Fig. 4: The legend needs to be graphically improved.

Yes.

I do not really see a real benefit of fig. 5. Maybe you can extract the results in a better way. One point might be that the model parameters are not explained and even the related processes are not highlighted in Fig. 5. In particular, it is not clear which information you can derive from the last place occurrence.

Please see my response to your comment 8.

It is not fully clear which information you can derived from investigating the most inferior process. It seems to be that this is either clear such as snowmelt parameter for California or related to the model structure.

The idea here is that modelers should not calibrate parameters associated with inferior processes in their watershed. If there are 35 calibration parameters, make sure to include the ones associated with the more dominate processes, and exclude the ones associated with the more inferior ones. I hope this idea comes across in the article.

Thank you for this reference list. I added citations to all of these references.

**Black text: S. Hoellering's comments**

**Red text: S. Markstrom's response**

**General comments**

The authors presented an interesting idea of a methodological framework wherein
parameters of the HRU based Precipitation-Runoff Modeling System (PRMS) can
be identified as influencial in terms of essential hydrological model based processes
and statistical streamflow indices serving as objective functions. Parameter influence
on model output was evaluated by parameter senstivity index values originating
from global sensitivity analysis with the Fourier Amplitude Sensitivity Test (FAST).
The approach aims at reducing the number of model input parameters to focus on
conceptualised processes assumed as hydrologically relevant within the watersheds
of the conterminous United States.

I generally agree with the concept of referencing model response functioning in
form of derived objective functions with dependent partial parameter sensitivities for
region specific model parameter identification. This is one of the aspects which would
be really worth publishing.
Apart from that, fundamental assumptions underlying this study are not sufficiently
clarified to address the discussed issues effectively, which are certainly
topical and relevant for model based catchment hydrology. The paper is technically
well-structured, exhibiting findings of the presented concept concisely but it lacks
the required presentation quality at too many different points. However, I found
some serious shortcomings and recommend to revise a number of major and minor
specific and technical points before the manuscript can be reconsidered for publication.

**Specific comments**
    1.   What is the main purpose of your paper?
You mention a number of issues e.g. "parameter identification", "process identification",
"calibration advise for modelers" or "identification of [model] structural
inadequacies". A better focus on one or two of these issues, preferably on the first
and second is advisable here.

Yes, the other review suggested that I discuss more the relationship between parameters and processes. I
think this is related to your comment here. I rewrote the introduction, with a focus on parameter and process
identification.

As uncertainty analysis is not the issue here, I furthermore suggest to remove the part starting from P16L29, which is also rather speculative.

Yes, that paragraph has been removed.

2. Please also name your assumptions more precisely!

The fundamental assumption of this study is, that the conceptualisation of PRMS

is structurally adequate to reproduce all hydrological processes of the CONUS. It is however not adressed, whether this assumption is valid or not or if the study doesn't claim to be transferable to real world processes and consequently stays a pure virtual

PRMS experiment. Conclusions on the dominant hydrological processes are only valid if it is shown that PRMS actually is a good representation of hydrological processes.

Processes in the study purely originate from and are defined by the PRMS structure whereby a comparison with observational data might be helpful in this application to show potential deficiencies or justify the fundamental assumption.

Yes, I restructured the PRMS methods section to include more about the calibration parameters and
assumption and less detail about how the application was set up.

3. P2L19/P10L20: As you similarly found out, more complex processes such as the reproduction of streamflow and its components as well as mountainous regions require more calibration parameters. The general rather small remaining subset of sensitive parameters explaining the majority of the model output variance of processes might be predefined by the conceptual structure of PRMS and a hint to overparameterization.

Yes, I added a sentence essentially saying this.

The number of parameters required in a process is also predetermined by the model/process concept and its complexity. Maybe be a bit more specific and less general or sketchy in stating your findings i.e. in the sense of the influence a parameter exerts on a process which might not be purely predetermined by the concept of a model.

Yes. Based on the suggestion of another reviewer, I have added another table (table 1) that lists the
parameters used in this study. In this table, I specify which "module type" each parameter is associated with
in the source code. So, without bogging down this article with too many model structure issues, maybe this
give the reader some idea of how the calibration parameters relate to the model structure.

4. P3L13: (How) do these two aspects of complexity correspond to the ones stated in the abstract and explained directly above these lines? Maybe you should be more precise here!

Yes. I added some text about using sensitivity analysis to reduce the complexity to the model user. That is my point. Obviously, SA does nothing about model structure, but the model can appear less complex to the modeler by focusing on those parameters and processes in the model that can be affected.

5. P3L32: This issue has also been partly discussed e.g. by Reusser and Zehe (2011).

Yes, added this reference.

5. P5L8: HRUs are purely derived and defined by their geographic and topographic location. Process identification and catchment classification might be hampered by this definition e.g. by mingling of processes leading to a complex interplay and location specific response behaviour which cannot be always captured by one HRU. In addition to your discussed points a redefinition of HRUs based on dominant hydrologic processes instead of the applied discretisation based on geographic position might be a conceivable outcome and a consequence of your study maybe helpful for calibration.

Yes, added to discussion section.

6. P5L20: Here a more precise explanation might be helpful. Is simulated streamflow at locations with stream gauges evaluated differently from streamflow at sites without observations?

I removed this sentence/section. The other reviewer felt this was too much detail about this aspect.

7. P7L1: Here more attention to further studies with streamflow indices could be given (see e.g. Yadav et al. (2007)). Please discuss your choice in some more details.

8. P9L25: I suggest to start this chapter with the sentence "To identify the expected count of parameters ... (P9L28)" first the theory, then a specific example.

Yes, I moved the text preceeding "To identify…" down to a subsequent summary paragraph.

9. P10L23/P13L8: This view might be kind of model structure/concept specific (as stated above) and is not surprising as streamflow is a convolution of these individual processes. Isn't total HRU runoff in PRMS the pure product or sum of the other streamflow processes (surface runoff, interflow and baseflow), hence involved process parameters add up to a larger number suggesting more complexity? Maybe you can be a bit more precise in the explanations (P13L13).

Yes, this is the point. Because process that happen "earlier" in the flow cycle affect the processes that happen later, there can be unexpected sensitivity of a process to a parameter that normally is not associated with that process. I added some text about this.

P15L25: To my knowledge PRMS offers different modules for PET calculations.
(How) do sensitivity results and parameter identificaton change by replacing one
module by another? This might be subject of future studies and worth mentioning.

Yes, added to "Further study" section.

P16L3: Someone who is interested in modelling the selected catchment is probably
better advised to have a look at historical meteorological observations. From
these it should be obvious that snowmelt might not be of any interest here.

Yes, that's an obvious one.

**Technical corrections**
Typing errors:
The spelling and writing needs improvement and proofreading. To mention several of
them:
Please be consistent in the writing and consider HESS manuscript preparation
guidelines for authors e.g. Figure, Fig.

Yes, fixed Table, Fig., and Figure.

P2L15: indicate instead of indicates

Yes, fixed.

P4L3/P16L14: watershed**s**

Yes, fixed.

P8L15: Here poor comprehensibility can be better avoided by changing three to
seven objective functions: "... 56 combinations of three objective functions and eight
processes (plus totals)."

Yes, fixed.

P11L7: "...**is** surprising..."

Yes, fixed.

P15L12: "This is probably because it **is** a major component of the hydrologic cycle that is..."

Yes, fixed.

P15L21: tha**n**

Yes, fixed.

P16L7: use**d**

Yes, fixed.

P16L11: "...**is** defined..."

Yes, fixed.

P16L14: process**es**

Yes, fixed.

Reference/citation errors:
Citations in the manuscript are correct while the year 2014 in complete reference is
not:
Markstrom, S. L., Regan, R. S., Hay, L. E., Viger, R. J., Webb, R. M. T., Payn,
R. A., and LaFontaine, J. H.: PRMS-IV, the precipitation-runoff modeling system,
version 4, U.S. Geological Survey Techniques and Methods, book 6, chap. B7, 158,
http://dx.doi.org/10.3133/tm6B7, **2015**

Yes, fixed.

Figures:
General remarks:
Resolution and quality of the presented figures and maps seem to be generally
not high enough or pixelated and need substantial improvement. Unfortunately, the
labeling of latitudinal and longitudinal lines are not readable at all. Please improve
the legibility or remove it or incorporate it in only one figure which might be enough to
show it once.

Yes, I have removed the lat/long lines from all maps. My original figure are of very much higher quality than
what is shown in the draft. MS Word seems to be importing them at a lower resolution than my originals. If this continues to be a problem, perhaps I can work with someone at HESS to ensure that the figures are high resolution.

Some of the shortcomings are listed here:

Figure 1: This map lacks both sufficient quality and a valuable information content.

In my oppinion a different form of presentation such as histograms or kernel density estimates for selected attributes of HRUs could be beneficial.

Yes, this figure has been removed.

Figure 2: Please use consistent spelling or abbreveations for objective functions across tables and figures. Please explain the additional column "Process average" in the results section 4.1 and the meaning of the legend.

The caption should also provide more information.

Yes, figure 2 has been remade with the same labels as table 2 (used to be table 1). I have also added a few sentences to explain "Process average" and how they are calculated.

Figure 3: Better use as Figure 1. It furthermore contains little information and poor legibility of region names.

Yes, this is now figure 1. I made the region labels larger.

Figure 4: "The plots A-**H** summarize..."

Yes, fixed.

Figure 5: Please clarify the connection to the ordered listing of Table 1.

Yes, added more to fig 4 (used to be figure 5) caption about this.

Figure 6: Please raise font sizes of titles above each map to be readable or remove them from the figure.

Yes, I removed them and remade the figures.

[revised manuscript text omitted]

Reusser and Zehe (2011). While, the user of a DPHM can do nothing about complexity associated with that model's internal structure, the apparent complexity to that user can be reduced by identifing those parameters and process that affect the DPHM in a particular application.

This article describes how this complexity can be addressed by focusing on parameter and hydrologic process identification through global parameter sensitivity analysis (SA). The

Commented [MSL4]: Guse:
P. 3, L. 6: The three references are related to studies which investigate performance measures more precisely. It might be good to also have a reference to studies which are directly investigating the model output.

Commented [MSL5]: decide if it is worthwile to abbreviate this.

degree to which different values of model parameters can affect the simulation of certain model outputs can be identified (G). Furthermore, parameter sensitivity can be evaluated with respect to selected output variables, each representing different aspects of the hydrologic cycle (hereafter refered to as "processes"). Sensitivity analysis of this form can be used to both identify the input parmeters that are the most sensitive (i.e. the parameters that affect the simulation the most) and the dominate process(es) (i.e. those processes which are affected most, by the most sensitive parameters).

Results of SA can vary spatially and must be accounted for as such. Specifically, DPHM parameters can be more or less sensitive at different locations on the landscape. For example, parameters related to simulation of snow can become more sensitive at higher elevations, while parameters related to evaporation can become less sensitive at locations where capacity for soil water storage decreases. Consequently, this means that the dominate process(es), as identified by SA, will vary across the landscape as well. These two issues are compounded as the spatial domain of the DPHM application expands. A common problem is that at large scale and with limited information, the effects of different hydrological processes can be indistinguishable from each other. For instance, groundwater recession and snowmelt from a receding snowpack can cause similar response in a streamflow hydrograph. If the prevailing hydrological process is not identified by the modeler, and subsequently parameterized in the model, the result can be "the right answer for the wrong reason" (Kirchner, 2006; McDonnell et al., 2007). This type of misunderstanding compounds both of the problems identified above as the modeler wastes resources working with insensitive input parameters and evaluating objective functions that do not relate with the real world physical processes. The result of these complex issues has led to study of parameter interaction (Clark and Vrugt, 2006) and equifinality (Beven, 2006).

Any particular DPHM must necessarily be complex because it must be able to simulate any and all hydrological process that may occur anywhere on the landscape. However, with the application of a DPHM to a specific site, it can become much less complex when the dominant hydrological process(es) are identified, as not all processes are active or at the same level of importance. The problem becomes less complex when hydrological processes not relevant to the modeled domain (or watershed) are removed from consideration (Wagener et al., 2003; Reusser et al., 2011; Guse et al., 2014; Bock et al., 2105; Bock et al., 2105). Dominant process concepts have been explored as a way to classify watersheds and natural hydrologic systems for simplifying DPHMs by several researchers (Sivakumar and Singh,

2012; Sivakumar et al., 2007).  Some have suggested the approach for use as a possible classification framework (e.g. Woods, 2002; Sivakumar, 2004).  Pfannerstill et al. (2015)

developed a framework for identification and verification of hydrologic process in simulation models on the basis of temporal sensitivity analysis. McDonnell et al. (2007) discuss the possibility of simplifying hydrologic modeling by identifying "fundamental laws" so that over parameterized models are not needed.  However, in our opinion we have not made much progress on that front and DPHMs are, in many ways and for many reasons, more complex than ever.

This article describes a SA for a modeling  approach application the conterminous United States (CONUS, Fig 1.).  The large domain is simulated by an  aggregated a  collection of many small scale watershed applications.

Identification and simulation of these watersheds is determined by the resolution of the available information and how the DPHM responds to geophysical (e.g., topography, vegetation and soils) and climatological variation.  Specifically, we propose to identify the sensitive parmaters and dominant hydrologic process(es), thereby identifying a reduced amount of  inputs and outputs to consider (Chaney et al., 2015).

**2  Methods**

**2.1  Distributed-parameter hydrology model **

The U.S. Geological Survey's  Precipitation-Runoff Modeling System (PRMS) is the

DPHM used in this study.  PRMS is a modular, deterministic, distributed-parameter, physical- process watershed model used to simulate and evaluate the effects of various combinations of

Commented [MSL6]: Hoellering: Rewrite the PRMS methods section to include more about PRMS parameters and assumption and less detail about how the application was set up.

1. Coordinate with Guse Methods comment 1: P. 4, L.29- P. 6, L.7: Please check carefully if you could reduce the subchapter 2.1 in length. Do you really need this information for this article?

[revised manuscript text omitted]

**Commented [MSL10]:** Hoellering: The caption should also provide more information.

[Figure]

Figure 3. Location Map of the conterminous United States showing the different geographic regions referred to this study.

[Figure]

Figure 43. Cumulative Precipitation-Runoff Modeling System parameter sensitivity across all HRUs in the continental Parameters Related to Processes. Parameter sensitivities have been averaged across all performance measures. The plots A- H summarize the counts for all 110,000 HRUs shown in the corresponding maps (I – P).

[Figure]

[Figure]

Parameter Occurrence

Figure 54. Frequency of occurrence of the different parameter counts. The count of circles in the row adjacent to the parameter name indicates how many times the respective parameter occurs in the different categories in table Table 12. The color of each circle indicates the ranking of that occurrence within the category, red corresponding to a higher ranking than blue.

Commented [MSL12]: Hoillering: Please clarify the connection to the ordered listing of Table 1.

[Figure]

[Figure]

[Figure]

Figure 65. Precipitation-Runoff Modeling System parameter sensitivity organized by process have been ranked for each hydrologic response unit for the entire conterminous United States (maps A and B) and for the Apalachicola – Chattahoochee – Flint River basin (maps C and D). The maps on the top (A and C) show the most dominate process, while the maps on the bottom (B and D) show the most inferior process.

---

## Referee Report (RR1)

**General comments**

The authors have submitted a revised version of 'Towards simplification of hydrologic modeling: identification of dominant processes'. In the manuscript a methodology of identification of influencial parameters and related dominant hydrological processes of the HRU based Precipitation-Runoff Modeling System (PRMS) is presented. Parameter influence on model output was evaluated by parameter senstivity index values originating from global sensitivity analysis with the Fourier Amplitude Sensitivity Test (FAST). The approach aims at reducing the number of calibration parameters helping modelers to focus on relevant processes within the watersheds of the conterminous United States.

The incorporation of necessary improvements in terms of the overall purpose, fundamental assumptions or the presentation quality of the study has been partly accomplished in a reasonable way. Nevertheless, in several parts, the manuscript still shows shortcomings. My concerns mainly relate to the structure and presentation of the concept, including the line of argumentation evolving around the general hypothesis of the study through the different sections. I recommend to revise again a number of specific and technical points to reach publication quality.

**Specific comments**

**Introduction**

I am not sure if the introduction is structured appropriately along the main purpose of the study defined as identification of 'sensitive parameters' and 'dominant processes'. I think it partly omits to set the right focus on these two aims which is important to understand the benefit of the methodology.

I like the beginning with the two complexities (input parameters and model output/processes) which nicely sets the focus on the main purpose and should structure the whole chapter, not to say the whole manuscript. Unfortunately, then the focus gets a bit lost and parts of this paragraph seem to be more a general description of methods (global sensitivity analysis, classification) and own findings which doesn't not keep this focus on the two complexities. Specifically, it is not sufficiently linked, first to previous studies, second to the presented study and results:

P3L1: Can you please be a bit more precise here in reference to the literature: What are reasons that parameters cannot be directly measured or transferred to larger scales even if measurements are partly available but at smaller scales (hillslope, plot or lab scale)?

P3L16: Are there any other studies where these two complexities (or one of them) were addressed or reduced?

P3L18: Here a few references (e.g. Sanadhya et al., 2013) to studies where global sensitivity analysis was used to identify parameters or processes might be worth to include.

P3L26-L31: References to these statements are missing - please add. Otherwise it can be regarded as one of your findings and sould be moved to the results or discussion sections. It seems to be an anticipation of your results or general interpretation of them.

P4L6: Could you be more specific here. How was the identification of dominant hydrological processes performed in some of these studies. I recommend to focus in more details on literature that has dealt with the identification of dominant parameters, hence processes. This is actually the purpose of your study and needs to include former efforts to cope with this complex problem. This should be presented to the reader in a form that clarifies the need for ongoing research on this topic, see e.g. Cuntz et al., 2015. Furthermore there might be studies where dominant processes are identified in different ways than with a purely model based approach.

P4L23: What kind of input do you mean (parameters or meteo forcing etc.?), output in which form? Can you please describe the input and output with a few words to be more clear here.

**Methods**

P4L30: Here it might be worth to mention the modular structure of PRMS first. This was first done in the following section 2.2. but is a property of PRMS.

P5L15: Are there studies where parameter interactions have been analysed.This might be additionally helpful to explain interactions of parameters.

P5L28: I suggest to delete the word 'stream segments' here to avoid confusion and focus on HRUs as fundamental spatial discretization units of PRMS. Stream segments where used to derive HRUs in the case of your study but are not further used for simulation and analysis.

P7L6: Please consider to rename the expression 'Performance measures'. In my view, performance measures are commonly used in hydrological modelling to evaluate the simulation results in comparison to any form of observation, which is not the case in this study. In accordance with B. Guse's comment on the previous manuscript version an expression should be selected that describes the statistical indices more appropriateley. I recommended to use a term like: 'fundamental daily streamflow statistics (FDSS)', 'statistical hydrological indices' or 'statistical response characteristics'.

P7L22: To be more consistent, I propose to move the section about the FAST analysis to the methods section. It is, in combination with the hydrological model and its output, a tool/method you used to identify parameters and processes.

P8L25: Please explain here or above why exactly a number of more than 9000 parameter sets are developped via FAST.

**Results**

P12L30: Could you better explain the connection of the circles' colors to the percentage values of Table 2 and to processes? (please see also comment on Fig. 4)

P13L24: Can you be a bit more precise here about sensitivity differences and the value for calibration between the two parameters and their order in vertical routing process in reference to the cited literature?

**Discussion**

P13L28, Section 5.1: In terms of the causes of parameter sensitivity, the discussion here is almost purely led from a model-based perspective without much reference to real world causes for dominant processes. I think it might be worth to structure it by model-based (as you already discussed ) and real world causes. I recommend to add here a small pararaph e.g. with an exemple of studies where dominant processes in the real world of the CONUS where identified and then relate it to your model based findings for causes of parameter sensitivity.

P16L1-P17L28, Section 5.3: Concerning the structure of this chapter and its role in the manuscript, in my opinion, parts of this chapter rather belong to the methods and results section. The authors first introduce a new procedure to make most dominant and inferior model based processes visible (P16L2-10) and then show its results in the following paragraphs. I recommend to split this section and move one part to the methods section, one to results and then discuss your findings in details in the section here.

P17L31: Isn't a calibration advise for modelers always one of the last outcomes of a research on hydrological modelling,

e.g. based on a feasible and sophisticated approach of parameter/process identification?

P19L3: ...or HRUs could be defined by dominant process instead of geographic location....

5  P19L3: 'Perhaps sensitivity analysis could help define this in a more objective way'. This statement seems to me very vague and should be formulated more clearly in relation to the previous sentence (see also comment directly above).

**Conclusion**

10  I have the impression that the conclusion is a bit vague and doesn't point at the most important and specific findings of the manuscript to a satisfactorily extent. Moreover, the two aspects of complexity stated in the introduction should be addressed here more specifically to reach a closed line of argumentation based on the main purpose/hypothesis of the study. The authors could also make use of the argumentation built on the two complexities to form the abstract, introduction and methods in a bit more consistent way. This structure should be kept also in order not to loose the readers attention in the different sections.

15
P19L13: As HRUs can be derived in various ways and their number is not fixed, I recommend to slightly change this sentence to something like: 'A global parameter sensitivity analysis was performed on the calibration parameters for all HRUs derived for the conterminous United States.'

20  **Technical corrections**

Please be consistent in the writing: '**dominate**' or '**dominant** process'. Please use the right adjective.

P2L13-L17: Is the order of findings listed here consistent with the order results are presented and discussed in the manuscript?
25  Please check and change if necessary.

*Typing errors:*

P2L11: I recommend to write: '...identify the (most) dominant process..'
30
P3L12: effect instead of affect

P5L29: are

35  *Tables and Figures:*

Table 1: In terms of the alphabetical sorting of parameters, I think it might be more useful to sort the parameters by PRMS (process) module according to their occurence in the vertical routing process. Additionally, another column showing the used value ranges of the calibration parameters could make sense in relation to the explanation of the FAST procedure on P8L23.

40
Figure 1: The resolution of the map still seems to be not high enough and labels a bit pixelated. The colors for the different elevation zones are faint and their contrast low.

Figure 3: The caption can be improved by stating that the plots and maps show the results for the different processes seperately.
45  In subplot (j) a vertical line is plotted to the right of the map. Please remove.

Figure 4/P12L30: It might be possible to assign percentage ranges or average percentages to the legend for first place occurences (red circles), in between occurence etc. to account for the degree of parameter influence throughout the CONUS. This would increase the information content of Fig. 4 and better illustrate the connection to Table 2. Furthermore, the figure doesn't

show any connection of the parameters to the processes they are influencing.

**References**

5   Cuntz, M., et al. (2015), Computationally inexpensive identification of noninformative model parameters by sequential screening, Water Resour. Res., 51, 64176441, doi:10.1002/2015WR016907.

Sanadhya, P., Giron J., and Arabi, M.: Global sensitivity analysis of hydrologic processes in major snow-dominated mountainous river basins in Colorado, Hydrological Processes, 28, 34043418, doi:10.1002/hyp, 2013.
10

---

## Author Response (AR2)

Suggestions for revision or reasons for rejection (will be published if the paper is accepted for final publication)

Review of the revised manuscript by Markstrom et al.

Guse's comments in black.

Markstrom's response in red.

I thank the authors for the careful revision of the manuscript and the consideration of my remarks. I think that the manuscript is now valid for publication after considering the minor remarks below.\

Thank you for your thoughtful and through review.

I encourage the authors to improve the readability of a couple of parts as suggested below including the abstract.

1. Page 2, Line 4-5: Unclear statement: „model output associated with dominate hydrological process(es)"

Yes, revised to "particular model output variables that could be associated with dominate hydrologic process (es)."

2. Page 2, Line 6-7: I did not understand the meaning of „on the basis of geographic location" in this context.

Yes. I was trying to convey that this analysis was done on the HRUs without the detail of defining/describing the HRUs in the abstract. I have revised the text to. "location on 110,000 independent hydrologically-based spatial modeling units covering the CONUS (HRUs)"

3. Page 2, Line 9-10: What do mean with „provide insights into model performance by location"?

Yes, revised to "provide insight into model performance at the location of each HRU"

4. Page 2, Line 27: Please check whether it should be written as „difficulty in the understanding…"

Yes, accepted.

5. Page 3, Line 29: The aspect that evaporation is not sensitive when the soil water storage is depleted is more a temporal aspect than a spatial, is it?

Yes, of course you are correct that the timing of water availability is what limits the ET. The point I was trying to make here is that ET becomes less sensitive to this timing if there is less capacity available. I have revised the text to "while parameters related to evaporation can become less sensitive at locations where soil depth and the overall capacity for soil water storage decreases." I hope this change reflects this idea better.

6. Page 7, Line 22: The FAST analysis is also a method. Thus, it is maybe more a chapter 2.5?

Yes, accepted. One of the coauthors thought that this section was enough of a standalone description to warrant its own top level section; however, both reviewers made this comments, so I moved it to "Methods" subsection.

7. Page 8, Line 25: Please add that the number of model results is provided by FAST and not selected subjectively.

Yes, I rewrote the section:

2.     Run the first part of the FAST procedure (as described above) to develop over 9000 unique parameter sets, comprised of value combinations for the calibration parameters. The total number and content of these parameter sets, and the results from their simulation by PRMS are completely determined by the first part of the FAST procedure in order to investigate the trial space. Each of the prescribed simulations are independent of each other so they can run in parallel on a computer cluster.

8. Page 9. Line 9-12: This means that you have summed up the first-order sensitivities for all parameters which are related to certain process? How did you realize this in the case that one parameter influence two processes, e.g. soil moisture and infiltration?

Yes. In this figure, I don't care if a certain parameter influences two processes or not. The question I'm trying to get at here is are there any parameters (I don't necessarily care which ones they are) that can be used to affect each category (combination of performance measure applied to output variable) of model output.

I revised the text to indicate that the sensitivities are summed for each process separately:

"In these maps, the HRUs are colored according to the parameter sensitivity, which is computed by summing the first-order sensitivity for all 35 parameters separately for each of the 8 output variables, each corresponding to their respective process. These sums do not necessarily sum to one…"

9. Page 10, Line 20: A clear definition of cumulative parameter sensitivity is missing. The meaning of this term is still not fully clear.

Yes. Up in the first paragraph of section 3.1 I added the sentence: "This summed sensitivity across the parameters, by each category is hereafter referred to as *cumulative parameter sensitivity*."

10. Page 10, Line 26: Why did you write „on average"

Yes, removed.

11. Page 10, Line 28: I do not think that it is useful to write that on average two parameters are required to represent snowmelt in the PRMS model. The map (Fig 3m) clearly shows the spatial heterogeneity and it becomes apparent that in the northern parts where snow is really relevant five to nine parameters are required. This spatial heterogeneity in the parameter count should be discussed as well for snowmelt (maybe included in the part on Page 11, Line 20-27).

Yes, I think your issue is with the word "average". I have removed that word, as the number of parameters does vary from 2 to 9.

12. Page 11, Line 5-11: This part is rather difficult to read. Could you maybe give more general statements instead of repeating the range of parameter counts for each process?

Yes, cut the repetitive parts. Not so tedious anymore.

13. Page 11, Line 14-16: It is somehow surprising that this part is not in the scope of the article. The statement „possibly indicate that some processes are overparametrized" is rather weak (and certainly not a results, but more a discussion part). However, I am surprised about this statement since the problem of overparameterization is highlighted in the introduction and it is remarked that progress in this topic is required (Page 4, Line 13-16). How does this match? Are you considering overparametrization or not? Concerning this, I would expect a clear statement.

Yes. I agree this section is not a result and I moved this down to the last paragraph in the "Further study" section. An in depth analysis of overparameterization in the model structure of PRMS is beyond the scope of this paper. This is really speculation as to why the parameters counts vary so much between the processes in this CONUS application.

14. Page 11, Line 28-31: For me, it is not clear how to extract from Fig. 3 the information that the parameter estimation is decomposed into separate problems. Here, an additional sentence would be helpful how to do this.

I added the sentence: "By considering a single (or reduced set of) process and performance measure categories at a time, the sensitive parameter space can be substantially reduced." I hope this is enough.

15. Page 12, Line 1-3: If not shown this statement is not helpful. At least, a figure/results in the supplementary would be required. Otherwise this part should be removed.

Yes, I joined two sentences together to make it clearer that the results from the analysis (that was not shown) is part of the procedure used to generate table 2.

16. Page 12, Line 11-19: Here, I recommend a short discussion of the relevance of this statement for hydrological modeling in general. I clearly becomes apparent that the impact and relevance of a performance measure even varies when considering separate processes. Thus, there is not only a relationship between processes and appropriate performance measures but also to the way how this process is adressed. I really makes a differences whether the timing or the total volume is considered. I think that this point should be emphasized even more, since it is a nice result and should be considered in future in calibration studies. Maybe you can emphasize this point even more in the discussion chapter 5.2.

Yes. Added a reiteration of this discussion to the "Choice of performance measure section" in the discussion section.

17. Fig 4: Does last place occurrence mean that this parameter is the last parameter among the sensitive parameters (as presented in Tab. 2) or even the least sensitive parameter at all? Maybe you can explain the meaning of last place occurrence.

Yes, I think some of the sentences were out of order in the figure caption. I have rewritten the caption. I hope this makes more sense.

18. Page 13, Line 17-18: I really like this concluding sentence. Maybe you can highlight it even more. At least, a new paragraph could be started after this sentence. I agree with the discussion later on Page 15, Lines 1-14.

Yes, I split the paragraph here. That does help to highlight this idea. This is strongly related to the step-by-step calibration strategy that we use to calibrate models. I.E. try to calibrate a particular parameter with information related to the first process in which it appears in the vertical routing order.

Hoellering's                    comments                    in                    black.
Markstrom's response is in red.

General comments:

The authors have submitted a revised version of 'Towards simplification of hydrologic modeling:
identification of dominant processes'. In the manuscript a methodology of identification of influencial
parameters and related dominant hydrological processes of the HRU based Precipitation-Runoff
Modeling System (PRMS) is presented. Parameter influence on model output was evaluated by
parameter senstivity index values originating from global sensitivity analysis with the Fourier Amplitude

Sensitivity Test (FAST). The approach aims at reducing the number of calibration parameters helping
modelers to focus on relevant processes within the watersheds of the conterminous United States.

The incorporation of necessary improvements in terms of the overall purpose, fundamental
assumptions or the presentation quality of the study has been partly accomplished in a reasonable
way. Nevertheless, in several parts, the manuscript still shows shortcomings. My concerns mainly
relate to the structure and presentation of the concept, including the line of argumentation evolving
around the general hypothesis of the study through the different sections. I recommend to revise again
a number of specific and technical points to reach publication quality.

Thank you for your thoughtful and thorough comments.

Specific comments

Introduction

I am not sure if the introduction is structured appropriately along the main purpose of the study
defined as identification of 'sensitive parameters' and 'dominant processes'. I think it partly omits to set
the right focus on these two aims which is important to understand the benefit of the methodology.

I like the beginning with the two complexities (input parameters and model output/processes) which
nicely sets the focus on the main purpose and should structure the whole chapter, not to say the
whole manuscript. Unfortunately, then the focus gets a bit lost and parts of this paragraph seem to be
more a general description of methods (global sensitivity analysis, classification) and own findings
which doesn't not keep this focus on the two complexities. Specifically, it is not sufficiently linked, first to previous studies, second to the presented study and results:

1. P3L1: Can you please be a bit more precise here in reference to the literature: What are
reasons that parameters cannot be directly measured or transferred to larger scales even if
measurements are partly available but at smaller scales (hillslope, plot or lab scale)?

Yes, added a bit of text and two references: Duan et al. (2005) describes "a gap in our
understanding of the links between model parameters and the land surface characteristics."
These unmeasured parameters, ostensibly tangible, are really empirical coefficients when it
comes to application and calibration (Samaniego et al., 2010).

2. P3L16: Are there any other studies where these two complexities (or one of them) were addressed or reduced?

Yes, added references to Jakeman and Hornberger, 1993; Hay et al., 2006

3. P3L18: Here a few references (e.g. Sanadhya et al., 2013) to studies where global sensitivity analysis was used to identify parameters or processes might be worth to include.

Yes, added this reference.

4. P3L26-L31: References to these statements are missing - please add. Otherwise it can be regarded as one of your findings and sould be moved to the results or discussion sections. It seems to be an anticipation of your results or general interpretation of them.

Yes, added reference to "van Werkhoven, K., Wagener, T., Reed, P., and Tang, Y.: Characterization of watershed model behavior across a hydroclimatic gradient, Water Resour. Res., 44, W01429, doi:10.1029/2007WR006271, 2008."

Another reviewer asked me to clarify the first sentences of this paragraph, so I added the example. Certainly this does anticipate the results. With the addition of the van Wekhoven reference I think it's OK to leave this in. My feeling is that it is not particularly novel to state that different watersheds across the landscape will have different parameter sensitivity.

5. P4L6: Could you be more specific here. How was the identification of dominant hydrological processes performed in some of these studies. I recommend to focus in more details on literature that has dealt with the identification of dominant parameters, hence processes. This is actually the purpose of your study and needs to include former efforts to cope with this complex problem. This should be presented to the reader in a form that clarifies the need for ongoing research on this topic, see e.g. Cuntz et al., 2015. Furthermore there might be studies where dominant processes are identified in different ways than with a purely model based approach.

Yes, added "Cuntz et al. (2015) describe a method of identifying only informative parameters as a screening step in order to reduce the effort required to perform global sensitivity analysis on the full parameter space. " to this paragraph.

Also added: "Various methods have been developed that will group similar catchments for purposes of study (Wolock et al., 2004; Winter, 2001; Ali et al., 2012) or for parameter regionalization (He et al., 2011; Merz and Blöschl, 2004, Seibert, 1999; Vogel 2005). "

6. P4L23: What kind of input do you mean (parameters or meteo forcing etc.?), output in which form? Can you please describe the input and output with a few words to be more clear here.

Yes, revised the last sentence of the introduction to: "Specifically, we propose to identify the sensitive parameters and dominant hydrologic process(es), thereby reducing the amount of parameter input and output variables to consider (Chaney et al., 2015) and address the two aspects of complexity as outlined above."

Methods

7. P4L30: Here it might be worth to mention the modular structure of PRMS first. This was first done in the following section 2.2. but is a property of PRMS.

Yes, changed the sentence to "Each hydrologic process simulated by the model PRMS is encoded in a modular piece of source code (i.e. a "module") and is represented within PRMS by an algorithm that is based on a physical law…"

8. P5L15: Are there studies where parameter interactions have been analysed.This might be additionally helpful to explain interactions of parameters.

Yes, I added several references. Some talk about issues related to parameters used in computations before other computations, others are more related to "parameter interaction".

9. P5L28: I suggest to delete the word 'stream segments' here to avoid confusion and focus on HRUs as fundamental spatial discretization units of PRMS. Stream segments where used to derive HRUs in the case of your study but are not further used for simulation and analysis.

Yes, deleted the words stream segment.

10. P7L6: Please consider to rename the expression 'Performance measures'. In my view, performance measures are commonly used in hydrological modelling to evaluate the simulation results in comparison to any form of observation, which is not the case in this study. In accordance with B. Guse's comment on the previous manuscript version an expression should be selected that describes the statistical indices more appropriateley. I recommended to use a term like: 'fundamental daily streamflow statistics (FDSS)', 'statistical hydrological indices' or 'statistical response characteristics'.

OK, how about "performance statistic"?

11. P7L22: To be more consistent, I propose to move the section about the FAST analysis to the methods section. It is, in combination with the hydrological model and its output, a tool/method you used to identify parameters and processes.

Yes.

12. P8L25: Please explain here or above why exactly a number of more than 9000 parameter sets are developped via FAST.

Yes, in response to the other reviewer, I rewrote: "Run the first part of the FAST procedure (as described above) to develop over 9000 unique parameter sets, comprised of value combinations for the calibration parameters. The total number and content of these parameter sets, and the results from their simulation by PRMS are completely determined by the first part of the FAST procedure in order to investigate the trial space. Each of the prescribed simulations…"

Results

13. P12L30: Could you better explain the connection of the circles' colors to the percentage values of Table 2 and to processes? (please see also comment on Fig. 4)

Yes, I reordered the sentences in the caption. This should reduce confusion. More discussion below on comment for Fig. 4.

14. P13L24: Can you be a bit more precise here about sensitivity differences and the value for calibration between the two parameters and their order in vertical routing process in reference to the cited literature?

Yes. I added the sentence: "In PRMS, the process of partitioning of precipitation into either direct surface runoff or infiltration (controlled directly by parameter *smidx_coef*) is "faster" and occurs in the vertical routing order before the process of interflow generation (controlled directly by parameter *slowcoef_sq*).

Discussion

15. P13L28, Section 5.1: In terms of the causes of parameter sensitivity, the discussion here is almost purely led from a modelbased perspective without much reference to real world causes for dominant processes. I think it might be worth to structure it by model-based (as you already discussed ) and real world causes. I recommend to add here a small pararaph e.g. with an exemple of studies where dominant processes in the real world of the CONUS where identified and then relate it to your model based findings for causes of parameter sensitivity.

I think your suggestion is a good idea, but I'm not sure how to do this. Other studies that I am aware of do not assign a dominant process category (e.g. "snowmelt" or " ET") in that same way that I present here. These studies use mathematical techniques like principle components, cluster analysis, regression, muiti dimensional distances, etc. to determine which HRUs or catchments are similar based on attributes. This is not necessarily the same thing as identifying the "dominant process."

There are also lots of examples of studies that result in maps of riparian areas, shallow water tables, etc. But, these are not really the same thing either.

16. P16L1-P17L28, Section 5.3: Concerning the structure of this chapter and its role in the manuscript, in my opinion, parts of this chapter rather belong to the methods and results section. The authors first introduce a new procedure to make most dominant and inferior model based processes visible (P16L2-10) and then show its results in the following paragraphs. I recommend to split this section and move one part to the methods section, one to results and then discuss your findings in details in the section here.

I don't think that any of this discussion belongs in the Methods section. The procedure outlined here is the analysis done on the sensitivity output, explaining how figure 5 was made. This is the same as figures 2 through 4.

I added an "Identification of dominant and inferior process by HRU" to the Results section and move most of the text from the Discussion to there.

I renamed the section in the discussion to "spatial aspect of dominant and inferior processes"

17. P17L31: Isn't a calibration advise for modelers always one of the last outcomes of a research on hydrological modelling, e.g. based on a feasible and sophisticated approach of parameter/process identification?

I would like to leave this sentence as the first in this paragraph as it defines "this approach" in the subsequent sentences.

18. P19L3: ...or HRUs could be defined by dominant process instead of geographic location....

Yes, changed to: "Also, alternative ways of defining HRUs (e.g. larger or smaller, or even based on dominant process instead of geographic location) could affect the analysis."

19. P19L3: 'Perhaps sensitivity analysis could 5 help define this in a more objective way'. This statement seems to me very vague and should be formulated more clearly in relation to the previous sentence (see also comment directly above).

I deleted the sentence. I think the previous sentence covers it.

Conclusion

I have the impression that the conclusion is a bit vague and doesn't point at the most important and specific findings of the manuscript to a satisfactorily extent. Moreover, the two aspects of complexity stated in the introduction should be addressed here more specifically to reach a closed line of argumentation based on the main purpose/hypothesis of the study. The authors could also make use of the argumentation built on the two complexities to form the abstract, introduction and methods in a bit more consistent way. This structure should be kept also in order not to loose the readers attention in the different sections.

Yes, added a paragraph about the two questions from the introduction.
20. P19L13: As HRUs can be derived in various ways and their number is not fixed, I recommend to slightly change this sentence to something like: 'A global parameter sensitivity analysis was performed on the calibration parameters for all HRUs derived for the conterminous United States.'

Yes.

Technical corrections

21. Please be consistent in the writing: 'dominate' or 'dominant process'. Please use the right adjective.

Yes.

22. P2L13-L17: Is the order of findings listed here consistent with the order results are presented and discussed in the manuscript? Please check and change if necessary.

Typing errors:

    23. P2L11: I recommend to write: '...identify the (most) dominant process..'

Yes.

    24. P3L12: effect instead of affect

Yes.

    25. P5L29: are

Tables and Figures:

    26. Table 1: In terms of the alphabetical sorting of parameters, I think it might be more useful to sort the parameters by PRMS (process) module according to their occurence in the vertical routing process. Additionally, another column showing the used value ranges of the calibration parameters could make sense in relation to the explanation of the FAST procedure on P8L23.

Yes. I also changed the order in section 2.3.

    27. Figure 1: The resolution of the map still seems to be not high enough and labels a bit pixelated. The colors for the different elevation zones are faint and their contrast low.

I've tried loading in "USA topo maps" from several sources in ArcMap and I get this same map every time. It is the USGS topo map for this resolution. I think it is the best map I can get of the landforms of CONUS.   Despite this, I made a new version of figure 1 that is 5 times higher resolution, specifying much larger text and halos. I don't think I can go much larger on the text without overlaps. I'm not sure that I can make this much better.

    28. Figure 3: The caption can be improved by stating that the plots and maps show the results for the different processes seperately. In subplot (j) a vertical line is plotted to the right of the map. Please remove.

Yes, I added "are shown by process" according to a comment by the other reviewer.

Yes, remade the figure without the line.

    29. Figure 4/P12L30: It might be possible to assign percentage ranges or average percentages to the legend for first place occurences (red circles), in between occurence etc. to account for the degree of parameter influence throughout the CONUS. This would increase the information content of Fig. 4 and better illustrate the connection to Table 2. Furthermore, the figure doesn't show any connection of the parameters to the processes they are influencing.

Yes, redid the figure showing percentages of HRUs that are affected.

References

[revised manuscript text omitted]